# Learning to Pool in Graph Neural Networks for Extrapolation

## Abstract

Graph neural networks (GNNs) are one of the most popular approaches to using deep learning on graph-structured data, and they have shown state-of-the-art performances on a variety of tasks. However, according to a recent study, a careful choice of pooling functions, which are used for the aggregation and readout operations in GNNs, is crucial for enabling GNNs to extrapolate. Without proper choices of pooling functions, which varies across tasks, GNNs completely fail to generalize to out-of-distribution data, while the number of possible choices grows exponentially with the number of layers. In this paper, we present GNP, a $L^p$ norm-like pooling function that is trainable end-to-end for any given task. Notably, GNP generalizes most of the widely-used pooling functions. We verify the effectiveness of GNNs on thirteen tasks using seven different GNN architectures and eight real-world graphs with up to one million edges. Specifically, we demonstrate that simply using GNP for every aggregation and readout operation enables GNNs to extrapolate well on many node-level, graph-level, set-related tasks; and GNP sometimes performs even better than the best-performing choices among existing pooling functions.

## 1 Introduction

Many real-world data, such as relationships between people in social networks or chemical bonds between atoms, can naturally be represented as graphs. Finding models with proper inductive biases to better describe such graph data has been a common goal for many researchers, and Graph Neural Networks (GNNs) (Scarselli et al., 2009; Kipf & Welling, 2017; Hamilton et al., 2017; Veličković et al., 2018; Xu et al., 2019; Maron et al., 2019; Xu et al., 2020) are considered to be the most successful model. They have proved effective for a variety of tasks, including recommendation (Ying et al., 2018a), drug discovery (Stokes et al., 2020), and chip design (Mirhoseini et al., 2020).

An important design choice for a GNN often overlooked is the specification of *pooling functions*, the functions used for the aggregation or readout operation in GNNs. They are usually required to be invariant w.r.t. the permutation of nodes in a graph, and common choices are element-wise summation (`sum`), maximum (`max`), minimum (`min`), or average (`mean`). Some recent works also proposed to use parametric models and learn them from data as well (Ying et al., 2018b; Lee et al., 2019b; Gao & Ji, 2019; Yuan & Ji, 2020).

While most of the previous works on this line focused on improving predictive performance for their own tasks, recently, Xu et al. (2021) studied the impact of the choice of pooling functions on the ability of a neural network to *extrapolate*. Specifically, Xu et al. (2021) highlighted the importance of the choice of pooling functions in order to make GNNs generalize over the data lying outside of the support of the training data distribution, and they argued that the specification of the pooling functions acts as an important inductive bias that can make GNNs either completely fail to extrapolate or gracefully generalize to out-of-distribution data. As a motivating example, consider the problem of counting the number of nodes in a graph. If we are to solve this problem with a single-layer GNN having one readout layer, probably the best pooling function would be `sum`, and the corresponding model will readily generalize to graphs with a much larger number of nodes than the ones seen during training. On the other hand, if we choose the pooling function as `max` instead, it may still fit the training data well but completely fail to predict the number of nodes in out-of-distribution graphs.

The findings in Xu et al. (2021) raise a natural question; which pooling functions should be used for a given problem in order to make GNNs constructed with them successfully extrapolate for

out-of-distribution data? Xu et al. (2021) did not present any guide but empirically showed that we do have the "right" pooling function for each problem tested, and when a pooling function is not properly selected, GNNs completely fails to extrapolate. The caveat here is that we do not know which pooling function is the right choice before actually training and validating the model.

To this end, in this paper, we present a generic learning-based method to find proper pooling functions for a given arbitrary problem. Our method, entitled Generalized Norm-based Pooling (GNP), formulates the pooling functions as a generic $L^p$ norm-like function (including negative $p$ as well), and learns the parameters inside the pooling functions in an end-to-end fashion. Unlike previous learning-based pooling methods that are usually tailored for specific tasks or focused on improving predictive performances, GNP can be applied to arbitrary tasks, and it improves the extrapolation ability of GNNs constructed with it. Also, GNP includes most of the pooling functions being used for GNNs as special cases. Despite the enhanced flexibility, GNP incurs minimal overhead in GNN in terms of the model complexity. A naïve application of GNP to GNNs is likely to fail because of some difficulty in training, so we propose a simple remedy to this. Using nine graph-level, node-level, and set-related tasks, we demonstrate that GNNs with GNP trained by our training scheme extrapolate for out-of-distribution data comparably and sometimes even better than those with pooling functions that are carefully chosen among of widely-used ones. In addition, we demonstrate the effectiveness of GNP on four real-world tasks (graph classification, influence maximization, node classification, and graph regression) using six additional GNN architectures (GCN (Kipf & Welling, 2017), GAT (Veličković et al., 2018), PNA (Corso et al., 2020), hierarchical SAGPool (Lee et al., 2019b), ASAPool (Ranjan et al., 2020), and MONSTOR (Ko et al., 2020)) and eight real-world graphs. We summarize our contributions as follows:

- **Generalized pooling function**: We propose GNP, a simple yet flexible pooling function that can readily be applied to arbitrary tasks involving GNNs, with minimal parameter overhead.
- **Effective training methods**: We propose effective training methods for GNP.
- **Extensive experiments**: We empirically demonstrate that GNNs with GNP generalize to out-of-distribution data on nine extrapolation tasks. We also show successful application of GNP to six GNN architectures on four real-world tasks in eight real-world graphs with up to one million edges.

## 2 RELATED WORK

**Aggregation functions**    Various aggregation functions have been appeared to enhance the performance of GNNs. Hamilton et al. (2017) proposed GraphSAGE with four different aggregation methods; `max`, `mean`, GCN (Kipf & Welling, 2017), and LSTM (Hochreiter & Schmidhuber, 1997). Veličković et al. (2018) proposed Graph Attention neTworks (GATs) including attention-based aggregation functions (Vaswani et al., 2017). Xu et al. (2019) proposed Graph Isomorphism Networks (GINs) and proved that GNN can satisfy the 1-Weisfeiler-Lehman (WL) condition only with `sum` pooling function as aggregation function. Recently, Li et al. (2020) proposed a trainable softmax and power-mean aggregation function that generalizes basic operators. Compared to these methods designed to improve interpolation performance on specific tasks, ours can improve extrapolation performance for generic tasks.

**Readout functions**    Zhang et al. (2018) suggested SortPooling that chooses top-$k$ values from the sorted list of the node features to construct outputs. Another popular idea is hierarchical pooling, where outputs are obtained by iteratively coarsening nodes in graphs in a hierarchical fashion (Ying et al., 2018b; Gao & Ji, 2019; Lee et al., 2019b; Yuan & Ji, 2020). Although demonstrated to be effective for the tasks they have been designed for, most of these methods require heavy computation and it is not straightforward to extend them for aggregation functions. On the other hand, our GNP can be applied to both aggregation and readout functions with minimal overhead.

**Pooling functions in generic context**    Vinyals et al. (2015) proposed Set2Set to get a representation of set-structured data with a LSTM-based pooling function. Lee et al. (2019a) proposed to use an attention-based pooling function to get summaries of set data. For convolutional neural networks, there were some approaches to generalize average pooling and max pooling widely used for many neural network architectures. Gulcehre et al. (2014) proposed a normalized learnable $L^p$ norm function that generalizes average pooling and max pooling. Lee et al. (2016) further extended those pooling functions with learnable tree-structured pooling filters.

**Norm-based pooling functions** There have been several works to employ norm-based pooling functions. Gulcehre et al. (2014) proposed a learnable $L^p$ norm function of the form

$$f(\mathbf{v}) = \left( \frac{1}{|\mathbf{v}|} \sum_{i=1}^{|\mathbf{v}|} |v_i|^p \right)^{1/p}$$

to substitute max pooling or average pooling used in convolutional neural networks. Similar norm-based pooling functions were used for acoustic modeling (Swietojanski & Renals, 2016) and text representation (Wu et al., 2020). Compared to GNP, these pooling methods cannot express the `sum` pooling. Li et al. (2020) further generalized this by multiplying $|\mathbf{v}|^q$ to include `sum` pooling as well, but not considered the case where $p$ is positive and the case where $p$ is negative at the same time. GNP is the most generic norm-based pooling function, compared to all aforementioned approaches, and more importantly, no other works studied their usefulness in the context of learning to extrapolate.

**Extrapolation** Trask et al. (2018) pointed out that most of the feed-forward neural networks fail to extrapolate even for the simplest possible identity mapping, and suggested using alternative computation units mimicking the behavior of arithmetic logic units. The ability to extrapolate is also important in the GNN context, for instance, many combinatorial optimization problems involving graphs often require extrapolation. Selsam et al. (2019); Prates et al. (2019) tackled the extrapolation problem by performing large iterations of message passing. Using various classical graph algorithms, Veličković et al. (2020) showed that the extrapolation performance of GNNs depends heavily on the choice of the aggregation function. Similarly, Xu et al. (2021) demonstrated that choosing the right non-linear function for both MLPs and GNNs is crucial for the extrapolation.

## 3 MAIN CONTRIBUTION: GENERALIZED NORM-BASED POOLING

In this section, we present our Generalized Norm-based Pooling (GNP) and discuss its expressiveness. Then, we describe some difficulties in training GNP and our remedy. Lastly, we present a task on which a GNN with GNP can extrapolate, while that equipped with the basic pooling functions cannot.

### 3.1 GENERALIZATION OF BASIC POOLING FUNCTIONS

While GNP is motivated by the $L^p$-norm function, which includes the `sum` and `max` functions as special cases, further ingredients are added to make GNP more flexible than the $L^p$-norm function. Specifically, we allow $p$ to be negative to let GNP express a wider class of functions than the previous norm-based or learning-based pooling functions.

Let $\mathbf{V} = \{\mathbf{v}_i\}_{i=1}^n$ be a set of node features with $\mathbf{v}_i \in \mathbb{R}^d$ for $i = 1, \ldots, n$. We define GNP to be an element-wise function where the output for each $j$th element is

$$\text{GNP}_j(\mathbf{V}) = \frac{1}{n^q} \left( \sum_{i=1}^n |v_{i,j}|^p \right)^{1/p},$$

where $p \in \mathbb{R} \setminus \{0\}$ and $q \in \mathbb{R}$ are learnable parameters. GNP includes the basic pooling functions (`sum`, `mean`, `max`, and `min`) as special cases.

**Proposition 1.** Suppose all the entries of $\mathbf{v}$ are non-negative in equation 3.1. Then, GNP includes `sum`, `max`, `mean` as special cases. If we further restrict $\mathbf{v}$ to be positive, GNP includes `min`.

*Proof.* $\text{GNP}_j(\mathbf{V})$ is equivalent to elementwise `sum` when $(p, q) = (1, 0)$ and elementwise `mean` when $(p, q) = (1, 1)$. When $q = 0$, we have

$$\lim_{p \to \infty} \text{GNP}_j(\mathbf{V}) = \max_i v_i \lim_{p \to \infty} \left( \sum_{i=1}^n \left( \frac{|v_{i,j}|}{\max_i |v_{i,j}|} \right)^p \right)^{1/p} = \max_i |v_{i,j}| \cdot 1 = \max_i |v_{i,j}|, \quad (1)$$

so GNP converges to `max`. Similarly, we can obtain `min` as a limit for $p \to -\infty$. □

## 3.2 HANDLING OF NEGATIVE $p$

The GNP function in equation 3.1 is not continuous and even not defined at $p = 0$. Hence, directly learning GNP in the original form as in equation 3.1 (even with $p = 0$ ignored) can cause instability, especially when an algorithm is trying to move from a positive $p$ value to a negative $p$ value. Instead, we suggest splitting the GNP function into two parts, $\text{GNP}^+$ with positive $p$ and $\text{GNP}^-$ with negative $p$, and let the model choose the right balance between them. Specifically, define

$$\text{GNP}_j^+(\mathbf{V}) = \frac{1}{n^{q^+}} \left( \sum_{i=1}^n |v_{i,j}|^{p^+} \right)^{1/p^+}, \quad \text{GNP}_j^-(\mathbf{V}) = \frac{1}{n^{q^-}} \left( \sum_{i=1}^n |v_{i,j}|^{-p^-} \right)^{-1/p^-}, \quad (2)$$

where $p^+ > 0$, $q^+$, $p^- > 0$, and $q^-$ are learnable parameters. Given a set of node features $\mathbf{V}$, we first split the feature dimension into two, and compute the output from $\text{GNP}^+$ for the first half and from $\text{GNP}^-$ for the second half. Then we mix two outputs with a single linear layer to get the final output.

$$\mathbf{y} = \begin{bmatrix} \text{GNP}_1^+(\mathbf{V}) & \dots & \text{GNP}_{\lfloor d/2 \rfloor}^+(\mathbf{V}) & \text{GNP}_{\lfloor d/2 \rfloor+1}^-(\mathbf{V}) & \dots & \text{GNP}_d^-(\mathbf{V}) \end{bmatrix}, \quad (3)$$

$$\text{GNP}(\mathbf{V}) = \mathbf{W}\mathbf{y} + \mathbf{b}, \quad (4)$$

where $\lfloor \cdot \rfloor$ is the floor function, $\mathbf{W} \in \mathbb{R}^{d \times d}$ and $\mathbf{b} \in \mathbb{R}^d$ are learnable parameters. Note that widely-used GNN layers have a linear layer or MLP after message-passing between nodes. Instead of using an additional linear layer, GNP concatenates the outputs of $\text{GNP}^+$ and $\text{GNP}^-$ and passes them to the linear layer or MLP. Therefore, we have only four extra parameters ($p^+$, $p^-$, $q^+$ and $q^-$) for each GNN layer of typical GNN architectures. With this design, GNP can easily switch between positive $p$ and negative $p$, choosing proper values according to tasks.

## 3.3 STABILIZATION OF TRAINING PROCESSES

Unfortunately, even with the above design to split the positive and negative parts, GNP still suffers from a training instability issue. In this section, we introduce our remedy for such an issue. With our remedy, as we will empirically demonstrate, GNP can be applied to arbitrarily complex deep GNNs as a drop-in replacement for the existing pooling functions.

**Negative or near-zero inputs**  GNP first processes inputs to be non-negative values by taking absolute values. In practice, in many GNN architectures, inputs are passed through ReLU before being fed into the pooling functions, so in such a case, we do not explicitly take the absolute values. If not, we explicitly put the ReLU activation function before every GNP to make inputs non-negative.

For the positive part of GNP, when the inputs are close to zero, the gradient w.r.t. the parameter $p^+$ may be exploded, as one can see from the following equation.

$$\frac{\partial \text{GNP}_j^+(\mathbf{V})}{\partial p^+} = \frac{\text{GNP}_j^+(\mathbf{V})}{p^+} \left( -\log(\text{GNP}_j^+(\mathbf{V})) + \frac{\sum_{i=1}^n v_{i,j}^{p^+} \log(v_{i,j})}{\text{GNP}_j^+(\mathbf{V})^{p^+}} \right).$$

Hence, we add a small tolerance term $\epsilon$ to every input element to prevent gradient explosion. This works well for positive $p$, but we need more care for negative $p$. When $p$ is negative, even small $\epsilon$ can be amplified by the term $(v_{i,j} + \epsilon)^{p^-}$ to dominate the other values. Hence, when a specific input $v_{i,j}$ is smaller than $\epsilon$, we replace it with $1/\epsilon$ to mask out the effect of that input for the output computation. An exceptional case is when every input element is below $\epsilon$. For such a case, we fix the output of GNP to be zero by default.

$$\tilde{v}_{i,j} = \begin{cases} v_{i,j} + \epsilon & \text{if } v_{i,j} > \epsilon \\ 1/\epsilon & \text{otherwise} \end{cases},$$

$$\text{GNP}_j^-(\tilde{\mathbf{V}}) = \begin{cases} 0 & \text{if } v_{i,j} < \varepsilon \text{ for } i = 1, \dots, n \\ \frac{1}{n^{q^-}} \left( \sum_{i=1}^n \tilde{v}_{i,j}^{p^-} \right)^{1/p^-} & \text{otherwise} \end{cases}.$$

Even with these treatments, still, the algorithm can diverge especially when $p$ is large. To resolve this, we clipped all $p$ values to be contained in $[0, 50]$ and used the log-sum-exp trick. That is,

$$\text{GNP}_j^{p^+}(\mathbf{V}) = \frac{1}{n^{q^+}} \exp\left( \frac{1}{p^+} \log\left( \sum_{i=1}^n \exp(p^+ \log(v_{i,j} + \epsilon)) \right) \right).$$

Table 1: Extrapolation performance in terms of MAPE on large graphs with different structures. On two tasks (`invsize` and `harmonic`), GNP significantly outperformed the second best one.

| Types | invsize | | harmonic | | maxdegree | |
|---|---|---|---|---|---|---|
| | GNP | Best Baseline (`sum`, `max`) | GNP | Best Baseline (SAGPool) | GNP | Best Baseline (`sum`, `max`) |
| BA | **0.9±0.3** | 92.5±10.5 | **2.5±0.9** | 78.4±40.8 | 2.1±1.1 | **0.0±0.0** |
| Expander | **1.9±1.0** | 35.4±7.8 | **0.9±0.5** | 11.9±18.6 | 2.3±1.1 | **0.0±0.0** |
| 4regular | **0.8±0.3** | 205.6±36.8 | **1.9±1.3** | 1179.3±310.6 | 3.4±3.7 | **0.0±0.0** |
| Tree | **0.8±0.3** | 202.3±11.9 | **14.7±6.3** | 149.4±34.9 | 1.9±0.6 | **0.0±0.0** |
| Ladder | **0.8±0.3** | 195.4±53.6 | **2.4±2.4** | 1138.4±283.3 | 30.7±16.9 | **0.1±0.1** |

Also, similar to Gulcehre et al. (2014), we reparameterized $p^+$ and $p^-$ with the softplus activation function, i.e., $p^+ = 1 + \log(1 + \exp(t^+))$ for some $t^+ \in \mathbb{R}$.

Another important trick was to use different learning rates for training $(p^+, p^-)$ and $(q^+, q^-)$. Since the parameters $(q^+, q^-)$ have much larger impact on the GNP, if we use the same learning rates for $(p^+, p^-)$ and $(q^+, q^-)$, the model can converge to unwanted local minimum that are not faithfully tuned for $(p^+, p^-)$. Hence, we used larger learning rates for $(p^+, p^-)$ to balance training.

### 3.4 Extrapolation Ability of GNP

As stated in Theorem 1, we prove that a GNN equipped with GNP can extrapolate on the `harmonic` task, which we define in Section 4.2. However, that equipped with the basic pooling functions cannot extrapolate on the task, as we show empirically in Section 4.2 and theoretically in Appendix A.

**Theorem 1.** (Informal) Assume all the nodes in $G$ have the same scalar feature 1. Then, a one-layer GNN equipped with GNP and trained with squared loss in the NTK regime learns the `harmonic` task function, and thus it can extrapolate.

*Proof.* See Appendix A for detailed analysis. □

## 4 Experiments

In this section, we review our experiments on various extrapolation tasks.

### 4.1 Experimental Setups

**Machines**    We performed all experiments on a Linux server with RTX 3090 GPUs.

**GNN models**    For graph-level tasks, we used one GIN (Xu et al., 2019) layer with a hidden dimension of 32 and two FC layers as MLP, and we fed only the outputs of the GIN layer into the readout function. Note that this simple model is expressive enough for obtaining exact answers to all considered graph-level tasks. For node-level tasks, we used three of the aforedescribed GIN layers, without readout functions, so that nodes at most three hops away from the target node can be taken into consideration. For set-level tasks, we used one FC layer with a hidden dimension of 32 before the pooling function and used another FC layer for the final output after the pooling function.

**Baseline**    Commonly for all tasks, we considered `sum`, `max`, `mean`, and `min`, all of which are generalized by GNP, as baseline aggregation and/or readout functions. For graph-level tasks, we additionally considered SortPooling (Zhang et al., 2018) with $k = 20$ and Set2Set (Vinyals et al., 2015) as baseline readout functions, and we considered the hierarchical pooling version of SAGPool (Lee et al., 2019b) as a whole as a baseline model. For set-level tasks, we additionally considered Set2Set (Vinyals et al., 2015) as a baseline pooling function and Set Transformer (Lee et al., 2019a) as a whole as a baseline model.

**Evaluation**    We compared evaluation metrics on the test set when validation loss was minimized, and in each setting, we reported mean and standard deviation over 5 runs, unless otherwise stated.

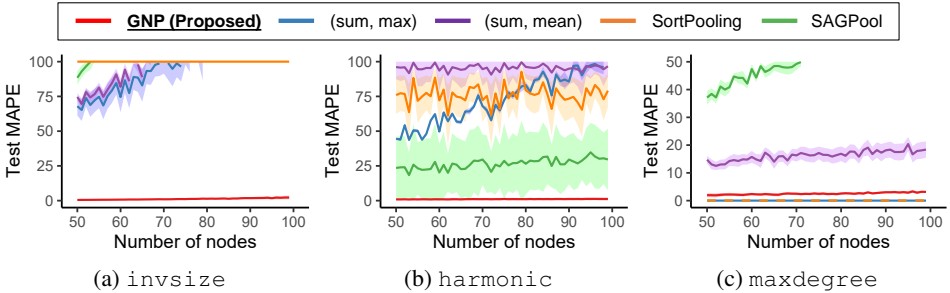

(a) `invsize`       (b) `harmonic`       (c) `maxdegree`

Figure 1: Extrapolation performances depending on the number of nodes in test graphs on three tasks (`invsize`, `harmonic`, and `maxdegree`). Only GIN equipped with GNP performed consistently well on all the tasks. We tested 19 competitors and reported the results of the most successful ones.

## 4.2 EXTRAPOLATION PERFORMANCES ON GRAPH-LEVEL TASKS

In this section, we consider three graph-level tasks. Given a graph, the first task is to find the maximum node degree (`maxdegree`), and the second task is to compute the harmonic mean node degree divided by the number of nodes (`harmonic`). The last task is to compute the inverse of the number of nodes (`invnode`), which does not depend on the topology of the given graph. For details of the synthetic datasets we used, see Appendix B.1.

For `maxdegree`, whose objective is $\max_{v \in V} \left( \sum_{u \in N(v)} 1 \right)$, where $N(v)$ is the set of neighbors of $v$, the reasonable choice is to use `sum` and `max` as aggregation and readout functions, respectively, For `harmonic`, whose objective is $\left( \sum_{v \in V} (\sum_{u \in N(v)} 1)^{-1} \right)^{-1}$ the reasonable combination of aggregation and readout functions are `sum` and GNP with $(p, q) = (-1, 0)$, respectively. For `invnode`, whose objective is $\left( \sum_{v \in V} 1^{-1} \right)^{-1}$, any of `mean`, `max`, and `min` is reasonable as the aggregation function, and GNP with $(p, q) = (-1, 0)$ is reasonable as the readout function.

We trained all models for 200 epochs, and we compared their test MAPE[1] for evaluation in Figure 1. GIN with GNP showed near-perfect extrapolation performances on all three tasks, and especially for `harmonic` and `invnode`, GIN with GNP was the only successful model. Among the 16 combinations of `sum`, `max`, `mean`, and `min`, using `sum` and `max` as the aggregation and readout functions, respectively, showed near-perfect extrapolation performance on `maxdegree`. For the same task, another combination (`mean`, `max`) showed reasonably good performance. For the other tasks, however, none of the 16 combinations was successful. SortPool and Set2Set as the readout function were tested, while fixing the aggregation function to the aforementioned reasonable one for each task. While they performed almost perfectly for `maxdegree`, they failed at the other tasks. Lastly, SAGPool was not successful in any of the tasks.

We also tested the extrapolation performance using large test graphs with distinctive structures. As seen in Table 1, GIN with GNP showed near-perfect performance only except for `harmony` on random trees, and `maxdegree` on ladder graphs. Especially, on `invsize` and `harmony`, it significantly outperformed the best baseline. We further tested the extrapolation performance of GNP and the baseline approaches using real-world graphs in Appendix C.1, graphs with different structures in Appendix C.2, graphs with different node feature distributions in Appendix C.3, and various activation functions in Appendix C.4.

## 4.3 EXTRAPOLATION PERFORMANCE ON NODE-LEVEL TASKS

We further evaluated the extrapolation performance of GNP on two node-level tasks considered in Veličković et al. (2020). The first task is to decide whether each node is within 3 hops from the target node or not. (`bfs`). We formulate the task as a regression problem and the label is 1 within 3 hops and 0 outside 3 hops. The second task is to find the minimum distance from each node to the target node on a graph with non-negative weights (`shortest`). Only the nodes within 3 hops from the target node were taken into consideration. As discussed in (Veličković et al., 2020), one of the optimal models for the tasks imitates the parallel breadth-first search and the parallel Bellman-Ford algorithm (Bellman, 1958) for `bfs` and `shortest`, respectively. In such cases, the reasonable aggregators for `bfs` and `shortest` are `max` and `min`, respectively.

---

[1]MAPE scales the error by the actual value, and it has been considered as a proper measure of extrapolation performance (Xu et al., 2021).

Table 2: Extrapolation performances in terms of MAE on two node-level tasks (`shortest` and `bfs`). GNP and all baseline methods were near perfect on `bfs`; GNP was second best on `shortest`.

| Aggregation | sum | max | mean | min | GNP |
|---|---|---|---|---|---|
| bfs | **0.000±0.000** | **0.000±0.000** | **0.000±0.000** | **0.000±0.000** | **0.000±0.001** |
| shortest | 1.323±0.162 | 0.762±0.395 | 1.316±0.330 | **0.141±0.007** | 0.332±0.105 |

(a) Extrapolation Performance on Large Graphs with Homogeneous Structures.

| Types | bfs | | shortest | | |
|---|---|---|---|---|---|
| | GNP | max | GNP | min | max |
| BA | 0.001±0.001 | **0.000±0.000** | 0.546±0.168 | **0.275±0.015** | 1.268±0.642 |
| Expander | **0.000±0.000** | **0.000±0.000** | 0.159±0.068 | **0.019±0.004** | 0.334±0.225 |
| 4regular | 0.003±0.003 | **0.000±0.000** | 1.911±0.257 | **1.188±0.182** | 5.178±1.218 |
| Tree | 0.003±0.002 | **0.000±0.000** | 1.579±0.289 | **1.057±0.256** | 4.584±0.980 |
| Ladder | 0.002±0.001 | **0.000±0.000** | 1.217±0.278 | **0.701±0.160** | 3.400±1.056 |

(b) Extrapolation Performance on Large Graphs with Heterogeneous Structures.

Table 3: Extrapolation performance in terms of MAPE on set-related tasks. Only the basic model equipped with GNP performed consistently well on all tasks. Especially for $\sigma^2_{\text{post}}$ and $\hat{\sigma}^2_{\text{MAP}}$, it significantly outperformed all competitors, including Set Transformer.

| Model | Pooling | $\mu_{\text{post}}$ | $\sigma^2_{\text{post}}$ | $\hat{\mu}_{\text{MAP}}$ | $\hat{\sigma}^2_{\text{MAP}}$ |
|---|---|---|---|---|---|
| Basic | sum | $135.0 \pm 9.3$ | $390.7 \pm 99.1$ | $126.8 \pm 18.2$ | $369.1 \pm 12.1$ |
| | max | $119.2 \pm 31.7$ | $120.6 \pm 4.0$ | $118.6 \pm 31.6$ | $108.8 \pm 2.0$ |
| | mean | $\underline{1.9 \pm 0.2}$ | $134.2 \pm 6.0$ | $\underline{1.9 \pm 0.2}$ | $107.1 \pm 2.2$ |
| | min | $95.8 \pm 15.6$ | $126.2 \pm 3.9$ | $118.6 \pm 31.6$ | $108.0 \pm 2.4$ |
| | Set2Set | $\underline{2.1 \pm 0.2}$ | $135.7 \pm 4.0$ | $\underline{1.9 \pm 0.2}$ | $106.1 \pm 2.6$ |
| Deep | sum | $136.3 \pm 6.8$ | $119.3 \pm 16.1$ | $100.0 \pm 0.0$ | $381.9 \pm 10.8$ |
| | max | $100.0 \pm 0.0$ | $123.8 \pm 2.1$ | $98.9 \pm 2.6$ | $109.6 \pm 3.3$ |
| | mean | $\underline{2.2 \pm 0.2}$ | $135.2 \pm 4.0$ | $\underline{2.2 \pm 0.4}$ | $109.2 \pm 2.9$ |
| | min | $83.0 \pm 10.2$ | $99.5 \pm 2.1$ | $90.8 \pm 6.0$ | $108.8 \pm 5.3$ |
| | Set2Set | $\underline{1.9 \pm 0.3}$ | $131.1 \pm 8.5$ | $\underline{1.9 \pm 0.2}$ | $106.0 \pm 1.5$ |
| Set Transformer | | $\underline{1.9 \pm 0.2}$ | $25.0 \pm 9.0$ | $\underline{1.9 \pm 0.1}$ | $40.8 \pm 9.5$ |
| **Basic** | **GNP** | $\mathbf{1.5 \pm 0.6}$ | $\mathbf{0.7 \pm 0.3}$ | $\mathbf{1.5 \pm 0.6}$ | $\mathbf{3.1 \pm 0.5}$ |

We considered five GINs equipped with `sum`, `max`, `mean`, and `min`, and GNP, respectively, as aggregation functions. Note that the readout operation is not used for node-level tasks. For description of the datasets, see Appendix B.2. We trained all of them for 100 epochs for `bfs` and for 200 epochs for `shortest`; and we compared their test MAE[2] in Table 2. GNP and all baseline methods were near perfect on `bfs`, regardless of graph types, and GNP was second best on `shortest`. As expected, GIN with `min` performed best on `shortest`.

## 4.4 EXTRAPOLATION PERFORMANCE ON SET-RELATED TASKS

We also applied our proposed approach to three set-related tasks. They are all related to estimating posterior distributions when the likelihood function is Gaussian. Specifically, the tasks are to find closed-form posterior hyperparameters $\mu_{\text{post}}$ and $\sigma^2_{\text{post}}$, the MAP estimate $\hat{\mu}_{\text{MAP}}$ of $\mu$ when $\sigma^2$ is known, and the MAP estimate $\hat{\sigma}^2_{\text{MAP}}$ if $\sigma^2$ when $\mu$ is known. Note that ground-truth values of $\mu_{\text{post}}$ and $\hat{\mu}_{\text{MAP}}$ are identical, while we used different loss functions for them. For description of the datasets, see Appendix B.3.

We trained for 300 epochs (a) the basic model (see Section 4.1) with GNP, (b) Set Transformer (Lee et al., 2019a) (c) the basic and deep[3] models with one among `sum`, `max`, `mean`, `min`, and Set2Set (Vinyals et al., 2015). We compared their MAPE in Table 3. The basic model equipped with GNP showed near-perfect extrapolation performance on all four tasks, even though the formula for $\hat{\sigma}^2_{\text{MAP}}$ cannot be exactly expressed by GNP, and it was the only such model. For $\mu_{\text{post}}$ and $\hat{\mu}_{\text{MAP}}$, whose ground-truth values are approximated by the average of the elements, Set Transformer and those equipped with `mean` or Set2Set were comparable to the basic model with GNP, while they were not on the other tasks.

---

[2]MAPE was not applicable since the ground-truth value for some nodes can be 0.

[3]The deep model has an additional FC layer before the pooling function.

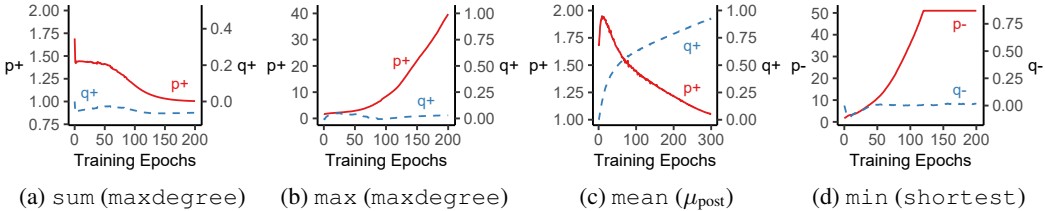

(a) sum (maxdegree)  (b) max (maxdegree)  (c) mean ($\mu_{\text{post}}$)  (d) min (shortest)

Figure 2: Empirical behavior of GNP. We showed how the parameters $p$ and $q$ of GNP changed during training. For each task, GNP imitated the ideal pooling functions if such pooling functions exist. For example, for maxdegree, GNP as aggregation and readout functions approximated sum (i.e., $p^+ \approx 1$ and $q^+ \approx 0$) and max (i.e., $p^+ \gg 1$ and $q^+ \approx 0$), respectively, which performed best.

Table 4: Effectiveness of GNP$^-$. The extrapolation performance of GNP degraded without GNP$^-$.

| Tasks | GNP | GNP$^+$ |
|---|---|---|
| harmonic | $\mathbf{1.1 \pm 0.8}$ | $2.1 \pm 0.6$ |
| shortest | $\mathbf{0.332 \pm 0.105}$ | $0.774 \pm 0.135$ |
| $\sigma^2_{\text{post}}$ | $\mathbf{0.7 \pm 0.3}$ | $0.6 \pm 0.2$ |

(a) Test Error on Erdős–Rényi Random Graphs

| Graphs | GNP | GNP$^+$ |
|---|---|---|
| BA | $\mathbf{2.5 \pm 0.9}$ | $31.5 \pm 1.2$ |
| tree | $\mathbf{14.7 \pm 6.3}$ | $26.1 \pm 7.4$ |
| ladder | $\mathbf{2.4 \pm 2.4}$ | $19.3 \pm 21.1$ |

(b) Test Error on harmonic on Other Graphs

## 4.5 EMPIRICAL BEHAVIOR OF GNP

As we discussed in Section 3, GNP generalizes sum, max, mean, and min. In order to confirm the facts experimentally, we showed in Figure 2 how the learnable parameters $p$ and $q$ in GNP changed during training. For maxdegree, GNP as aggregation and readout functions approximated sum (i.e., $p^+ \approx 1$ and $q^+ \approx 0$) and max (i.e., $p^+ \gg 1$ and $q^+ \approx 0$), respectively, which performed best on the task. For $\mu_{\text{post}}$ and shortest, GNP approximated mean (i.e., $p^+ \approx 1$ and $q^+ \approx 1$) and min (i.e., $p^- \gg 0$ and $q^- \approx 0$), respectively, which were the best performing baseline for the tasks. To sum up, empirically, GNP imitated the ideal pooling functions for each task if such pooling functions exist. We also observed that either GNP$^+$ or GNP$^-$ tends to dominate the other side in all considered graph-level tasks. Detailed results are provided in Appendix C.6.

## 4.6 ABLATION STUDY: EFFECTIVENESS OF GNP$^-$

In order to demonstrate the effectiveness of GNP$^-$ for extrapolation, we compared the model equipped only with GNP and the model only with GNP$^+$ on each of three tasks (harmony, shortest, and $\sigma^2_{\text{post}}$) in Table 4. The detailed settings for each task were the same as in previous experiments. The model only with GNP$^+$ performed well only on the task for $\sigma^2_{\text{post}}$. The extrapolation performance of GNP degraded significantly without GNP$^-$ on harmony and shortest.

## 4.7 EFFECTIVENESS OF GNP ON REAL-WORLD TASKS

**Graph classification** We compared the graph classification accuracy of hierarchical SAGPool (Lee et al., 2019b) and ASAPool (Ranjan et al., 2020), and their variants with GNP. For the variant of SAGPool, we replaced all pooling functions before, inside, and between graph pooling operations. For the variant of ASAPool, we replaced all pooling functions except for those inside LEConv. Since we used GNP, instead of the concatenation of global average pooling and max pooling functions, the input dimension of the first fully-connected layer after them was reduced by half. For the variants, except for the additional hyperparameters of GNP, all hyperparemters were set the same.

We used three datasets from TUDataset (Morris et al., 2020). D&D (Dobson & Doig, 2003; Shervashidze et al., 2011) and PROTEINS (Dobson & Doig, 2003; Borgwardt et al., 2005) contain protein-interaction graphs, and NCI1 (Wale & Karypis, 2006) contains the graphs representing chemical compounds. For consistency with the original SAGPool, we performed 10-fold cross validation with 20 different random seeds. For ASAPool, we performed 10-fold cross validation with the 20 random seeds specified in its implementation. We report the test accuracy with standard deviation in Table 5. SAGPool and ASAPool equipped with GNP consistently outperformed the original models with a carefully chosen pooling functions.

Table 5: Graph classification accuracy. Replacing the carefully chosen pooling functions in SAGPool and ASAPool with GNP improved their accuracy on graph-classification tasks.

| Model | Aggregation | Readout | D&D | PROTEINS | NCI1 |
|---|---|---|---|---|---|
| SAGPool (original) | GCN | mean, max | $0.765 \pm 0.009$ | $0.722 \pm 0.008$ | $0.688 \pm 0.013$ |
| SAGPool (with GNP) | **GNP** | **GNP** | $\mathbf{0.774 \pm 0.010}$ | $\mathbf{0.728 \pm 0.013}$ | $\mathbf{0.695 \pm 0.015}$ |

(a) SAGPool

| Model | Aggregation | Readout | D&D | PROTEINS | NCI1 |
|---|---|---|---|---|---|
| ASAPool (original) | GCN | mean, max | $0.764 \pm 0.009$ | $0.738 \pm 0.008$ | $0.711 \pm 0.004$ |
| ASAPool (with GNP) | **GNP** | **GNP** | $\mathbf{0.772 \pm 0.007}$ | $\mathbf{0.739 \pm 0.006}$ | $\mathbf{0.725 \pm 0.007}$ |

(b) ASAPool

Table 6: Influence maximization performance. The influences of 100 seed nodes produced by MONSTOR and its variants in graphs unseen during training are reported. The variant equipped with GNP outperforms original MONSTOR (with max) and the other variant (with sum).

| Aggregation | Extended | | | WannaCry | | | Celebrity | | |
|---|---|---|---|---|---|---|---|---|---|
| | BT | JI | LP | BT | JI | LP | BT | JI | LP |
| max | $1222.5\pm0.4$ | $706.9\pm0.1$ | $3259.6\pm0.7$ | $2746.5\pm1.4$ | $1646.6\pm2.1$ | $9090.2\pm3.8$ | $155.2\pm0.1$ | $\mathbf{140.5\pm0.0}$ | $5665.0\pm1.4$ |
| sum | $1216.6\pm1.7$ | $706.5\pm0.2$ | $3189.2\pm6.9$ | $2742.6\pm0.9$ | $1645.8\pm0.2$ | $9030.1\pm2.0$ | $153.9\pm0.4$ | $\mathbf{140.5\pm0.0}$ | $\mathbf{5666.9\pm0.6}$ |
| **GNP** | $\mathbf{1223.0\pm0.3}$ | $\mathbf{707.3\pm0.2}$ | $\mathbf{3262.1\pm1.7}$ | $\mathbf{2753.4\pm0.1}$ | $\mathbf{1648.3\pm0.1}$ | $\mathbf{9098.4\pm2.2}$ | $\mathbf{155.3\pm0.8}$ | $140.4\pm0.0$ | $5666.1\pm1.8$ |

**Influence maximization** We compared the performance of MONSTOR (Ko et al., 2020) and its variants with GNP on the influence maximization task (Kempe et al., 2003), which has been extensively studied due to its practical applications in viral marketing and computational epidemiology. The objective of the task is to choose a given number of seed nodes so that their collective influence (i.e., degree of spread of information through a given social network) is maximized.

For experimental details, we followed (Ko et al., 2020): (a) we used three real-world social networks with up to $85,202$ edges (Extended, WannaCry, and Celebrity) with three kinds of realistic activation probabilities (BT, JI, and LP), (b) we used the same training methods and hyperparameters except for the additional parameters of GNP, and (c) we compared MONSTOR and its variants in an inductive setting. For example, we used the model trained using the Celebrity and WannaCry datasets to test the performance on the Extended dataset. For details of the influence maximization problem, MONSTOR, and the real-world social networks that we used, see Appendix E.

We performed three runs and reported the influence maximization performance with standard deviations in Table 6. As seen in the results with sum and max aggregations, the performances heavily depended on the choice of the aggregation function. In most of the cases, MONSTOR equipped with GNP outperformed the original MONSTOR with max aggregation and also a variant of MONSTOR with sum aggregation.

**Two additional tasks** The experimental results on node classification and graph regression are provided in Appendix C.7. For node classification, we used a real-world graph with over one million edges and compared the accuracy of GCN and GAT, and their variants with GNP. For graph regression, we compared the MAE of PNA (Corso et al., 2020) and its variants with GNP.

## 5 CONCLUSION

In this work, we proposed GNP, a learnable norm-based pooling function that can readily be applied to arbitrary GNNs or virtually to any neural network architecture involving permutation-invariant pooling operation. The key advantages of GNP are its generality and ability to extrapolate. We showed that GNP includes most of the existing pooling functions and can express a broad class of pooling functions as its special cases. More importantly, with various synthetic and real-world problems involving graphs and sets, we demonstrated that the networks with GNP as aggregation or readout functions can correctly identify the pooling functions that can successfully extrapolate. We also introduced some non-trivial design choices and techniques to stably train GNP. The limitation of our work is that, although we have empirically demonstrated the excellent extrapolation performance on various tasks, we have not developed theoretical arguments regarding under what condition models constructed with GNP will extrapolate well. It would be an interesting future work to rigorously study the class of problems that GNP can solve.

REPRODUCIBILITY STATEMENT

We provided the source code used in our experiments in main paper, including the implementations of GNP and the GIN model, in the supplementary materials. The provided supplementary matarials also include example synthetic datasets and the pretrained weights used in our experiments.

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

## A  THEORETICAL ANALYSIS (RELATED TO SECTION 3.4)

Similarly to Xu et al. (2021), we present an analysis of the extrapolation ability of GNNs with our GNP pooling function. Specifically, we show that a one-layer GNN with the GNP pooling function can extrapolate on the `harmonic` task. Let $f(\boldsymbol{\theta}, G)$ be a one-layer GNN defined as follows:

$$f(\boldsymbol{\theta}, G) = W^{(2)} \operatorname{GNP}\left(\left\{\sum_{v \in N(u)} W^{(1)} \boldsymbol{x_v}\right\}_{u \in V}\right), \tag{5}$$

where $\boldsymbol{\theta}$ is the parameters of the GNN, $G = (V, E)$ is the input graph, $x_v$ is the initial feature of node $v \in V$, and $N(u) \subseteq V$ is the set of the neighbors of node $u \in V$.

When two graphs $G$ and $G'$ are given, the Graph Neural Tangent Kernel (GNTK) (Du et al., 2019) is computed as

$$\operatorname{GNTK}(G, G') = \mathbb{E}_{\boldsymbol{\theta} \sim \mathcal{N}(0, I)}\left[\left\langle \frac{\partial f(\boldsymbol{\theta}, G)}{\partial \boldsymbol{\theta}}, \frac{\partial f(\boldsymbol{\theta}, G')}{\partial \boldsymbol{\theta}} \right\rangle\right].$$

### A.1  FEATURE MAP OF THE GNTK

We first compute the GNTK for the network defined as equation 5 and derive the corresponding feature map using a general framework presented in Jacot et al. (2018); Du et al. (2019); Xu et al. (2021). Let $\Sigma^{(1)}, \Sigma^{(2)}$ be the covariance for the first linear layer and the second linear layer, respectively. Also, let $\Theta^{(1)}$ be the kernel value after the first linear layer, respectively.

From the framework, $\Sigma^{(1)}$ and $\Theta^{(1)}$ are determined as follows:

$$\left[\Sigma^{(1)}(G, G')\right]_{uu'} = \left[\Theta^{(1)}(G, G')\right]_{uu'} = \boldsymbol{x}_u^\top \boldsymbol{x}_u.$$

Also, $\Sigma^{(2)}$ can be computed as

$$\Sigma^{(2)}(G, G') = \mathbb{E}_{(f(v), f(v')) \sim \mathcal{N}\left(\mathbf{0}, [\Lambda^{(1)}(G, G')]_{vv'}\right)}\left[\operatorname{GNP}\left(\left\{\sum_{v \in N(u)} f(v)\right\}_{u \in V}\right)\right.$$
$$\left.\operatorname{GNP}\left(\left\{\sum_{v' \in N(u')} f(v')\right\}_{u' \in V'}\right)\right],$$

where

$$[\Lambda^{(1)}(G, G')]_{vv'} = \begin{pmatrix} [\Sigma^{(1)}(G, G)]_{vv} & [\Sigma^{(1)}(G, G')]_{vv'} \\ [\Sigma^{(1)}(G', G)]_{v'v} & [\Sigma^{(1)}(G', G')]_{v'v'} \end{pmatrix} = \begin{pmatrix} \boldsymbol{x}_v^\top \boldsymbol{x}_v & \boldsymbol{x}_v^\top \boldsymbol{x}_v' \\ \boldsymbol{x}_v'^\top \boldsymbol{x}_v & \boldsymbol{x}_v'^\top \boldsymbol{x}_v' \end{pmatrix}.$$

By a simple algebraic manipulation, one can easily see that the feature map $\phi(G)$ is computed as

$$\phi(G) = c \cdot \left(\operatorname{GNP}\left(\{\boldsymbol{w}^{(k)^\top} \boldsymbol{h_v}\}_{v \in V}\right),\right.$$
$$\left.\frac{1}{|V|^q} \sum_{u \in V} (\boldsymbol{w}^{(k)^\top} \boldsymbol{h_u})^{p-1} \left(\sum_{v \in V} (\boldsymbol{w}^{(k)^\top} \boldsymbol{h_v})^p\right)^{\frac{1}{p}-1} \mathbb{I}(\boldsymbol{w}^{(k)^\top} \boldsymbol{h_u} > 0) \cdot \boldsymbol{h_u}, \dots\right), \tag{6}$$

where $\boldsymbol{w}^{(k)} \sim \mathcal{N}(\boldsymbol{0}, \boldsymbol{I})$, $c$ is a constant, and $\boldsymbol{h}_u$ is the sum of the initial features of the neighbors $N(u)$ of node $u$, i.e. $\sum_{v \in N(u)} \boldsymbol{x}_v$.

## A.2 ANALYSIS ON THE `HARMONIC` TASK

We analyze the extrapolation ability of GNNs on the `harmonic` task, whose exact functional form is given as

$$f^{\star}(G) = \left( \sum_{v \in V} \left( \sum_{u \in N(v)} 1 \right)^{-1} \right)^{-1}. \tag{7}$$

Following Xu et al. (2021), we assume *linear algorithmic alignment*; if a neural network can simulate a target function $f$ by replacing MLP modules with *linear* functions, (i.e., the nonlinearities of the neural network is well matches with the target function, so the neural network only has to learn the linear (MLP) part), than it can correctly learn the target function, and thus can extrapolate well. With this hypothesis, we proceed as follows. We assume that a GNN is in the NTK regime, that is, the GNN is initialized in a specific way called NTK parameterization, trained via gradient descent with small step size, and the widths of the network tend to infinty. In such case, the GNN behaves as a solution to kernel regression with GNTK kernel. Then we convert the kernel regression problem into a constrained optimization problem in the feature space induced from GNTK kernel, and show that the solution for the constrained optimization problem aligns with the functional form of the `harmonic` task (equation 7).

We first state the following Lemma from Xu et al. (2021) showing that a NTK kernel regression solution can be viewed as a constrained optimization problem in the feature space.

**Lemma 1 (Lemma 2 in Xu et al. (2021)).** Suppose $\text{NTK}_{\text{train}}$ is the $n \times n$ kernel for training data, $\text{NTK}(\boldsymbol{x}, \boldsymbol{x}_i)$ is the kernel value between test data $\boldsymbol{x}$ and training data $\boldsymbol{x}_i$, and $Y$ is the training labels. Let $\phi(\boldsymbol{x})$ be a feature map induced by a neural tangent kernel, for any $x \in \mathbb{R}^d$. The solution to kernel regression

$$(\text{NTK}(\boldsymbol{x}, \boldsymbol{x}_1), \ldots, \text{NTK}(\boldsymbol{x}, \boldsymbol{x}_n)) \cdot \text{NTK}_{\text{train}}^{-1} Y$$

is equivalent to $\phi(\boldsymbol{x})^{\top} \boldsymbol{\beta}_{\text{NTK}}$, where $\boldsymbol{\beta}_{\text{NTK}}$ is

$$\min_{\boldsymbol{\beta}} \|\boldsymbol{\beta}\|_2$$
$$\text{s.t. } \phi(\boldsymbol{x}_i)^{\top} \boldsymbol{\beta} = y_i, \quad \text{for } i = 1, \ldots, n. \tag{8}$$

*Proof.* See Xu et al. (2021). □

**Theorem 1.** Assume all the nodes in $G$ have the same scalar feature 1. Then, a GNN defined as equation 5 trained with squared loss in the NTK regime learns the `harmonic` task function (equation 7).

*Proof.* Assume $(p^+, p^-, q^+, q^-) = (\infty, 1, \infty, 0)$. Then every output of $\text{GNP}^+$ goes to zero regardless of inputs and $\text{GNP}^-$ aligns with the target function equation 7. The feature map of GNTK in this case can be simplified as follows:

$$\phi(G) = c \cdot \left( \text{GNP}\left( \{ \boldsymbol{w}^{(k)^{\top}} \boldsymbol{h}_{\boldsymbol{v}} \}_{v \in V} \right), \right.$$

$$\left. \sum_{u \in V} (\boldsymbol{w}^{(k)^{\top}} \boldsymbol{h}_{\boldsymbol{u}})^{-2} \left( \sum_{v \in V} \frac{1}{\boldsymbol{w}^{(k)^{\top}} \boldsymbol{h}_{\boldsymbol{v}}} \right)^{-2} \mathbb{I}(\boldsymbol{w}^{(k)^{\top}} \boldsymbol{h}_{\boldsymbol{u}} > 0) \cdot \boldsymbol{h}_{\boldsymbol{u}}, \ldots \right).$$

By **Lemma 1**, we know that in the NTK regime, the GNN $f(\boldsymbol{\theta}, G)$ behaves as the solution to the constrained optimization problem equation 8 with feature map $\phi(G)$ and coefficients $\boldsymbol{\beta}$. Let $\hat{\boldsymbol{\beta}}_{\boldsymbol{w}} \in \mathbb{R}$ be a coefficient corresponding to $\text{GNP}(\{\boldsymbol{w}^{\top} \boldsymbol{h}_u\}_{u \in V})$ and $\hat{\boldsymbol{\beta}}'_{\boldsymbol{w}} \in \mathbb{R}$ be a coefficient corresponding to the other term in $\phi(G)$. Similar to Lemma 3 in Xu et al. (2021), we can combine the effect of coefficients for $\boldsymbol{w}$'s in the same direction. For each $\boldsymbol{w} \sim \text{Unif}(\text{unit sphere})$, we can define $\boldsymbol{\beta}_{\boldsymbol{w}}$ and $\boldsymbol{\beta}'_{\boldsymbol{w}}$ as the total effect of weights in the same direction with considering scaling.

$$\beta_{\boldsymbol{w}} = \int \beta_{\boldsymbol{u}} \mathbb{I}\left(\frac{\boldsymbol{w}^{\top}\boldsymbol{u}}{\|\boldsymbol{w}\|\|\boldsymbol{u}\|} = 1\right) \cdot \frac{\|\boldsymbol{u}\|}{\|\boldsymbol{w}\|} \mathbb{P}(\boldsymbol{u}),$$

$$\beta'_{\boldsymbol{w}} = \int \beta'_{\boldsymbol{u}} \mathbb{I}\left(\frac{\boldsymbol{w}^{\top}\boldsymbol{u}}{\|\boldsymbol{w}\|\|\boldsymbol{u}\|} = 1\right) \cdot \frac{\|\boldsymbol{u}\|}{\|\boldsymbol{w}\|} \mathbb{P}(\boldsymbol{u}).$$

Since the dimension of the input features is 1, we only need to consider two directions of $\boldsymbol{w}$. To get min-norm solution, we compute the Lagrange multiplier as

$$\min_{\hat{\beta},\hat{\beta}'} \int \hat{\beta}_{\boldsymbol{w}}^2 + \hat{\beta}_{\boldsymbol{w}}'^2 d\mathbb{P}(\boldsymbol{w})$$

$$s.t. \int \mathrm{GNP}\left(\{\boldsymbol{w}^{\top}\boldsymbol{h}_{\boldsymbol{v}}\}_{v \in V}\right) \cdot \hat{\boldsymbol{\beta}}_{\boldsymbol{w}} + \sum_{u \in V}(\boldsymbol{w}^{\top}\boldsymbol{h}_{\boldsymbol{u}})^{-2}\left(\sum_{v \in V}\frac{1}{\boldsymbol{w}^{\top}\boldsymbol{h}_{\boldsymbol{v}}}\right)^{-2}\mathbb{I}(\boldsymbol{w}^{\top}\boldsymbol{h}_{\boldsymbol{u}} > 0) \cdot \hat{\boldsymbol{\beta}}'_{\boldsymbol{w}} \cdot \boldsymbol{h}_{\boldsymbol{u}} d\mathbb{P}(\boldsymbol{w})$$

$$= \left(\sum_{u \in V_i}\boldsymbol{h}_u^{-1}\right)^{-1} \quad \forall i \in [n],$$

where $G_i = (V_i, E_i)$ is the $i$-th training data and $\boldsymbol{w} \sim \mathcal{N}(0,1)$. By KKT condition, taking the derivative for each variable, we can get the following conditions:

$$\hat{\boldsymbol{\beta}}_+ = c \cdot \sum_{i=1}^n \lambda_i \cdot \left(\sum_{u \in V_i}\boldsymbol{h}_u^{-1}\right)^{-1},$$

$$\hat{\boldsymbol{\beta}}_- = 0$$

$$\hat{\boldsymbol{\beta}}'_+ = c \cdot \sum_{i=1}^n \lambda_i \cdot \left(\sum_{u \in V_i}\boldsymbol{h}_u^{-1}\right) \cdot \left(\sum_{v \in V_i}\boldsymbol{h}_v^{-1}\right)^{-2}$$

$$= c \cdot \sum_{i=1}^n \lambda_i \cdot \left(\sum_{u \in V_i}\boldsymbol{h}_u^{-1}\right)^{-1},$$

$$\hat{\boldsymbol{\beta}}'_- = 0,$$

$$\left(\sum_{u \in V_i}\boldsymbol{h}_u^{-1}\right)^{-1} = \hat{\boldsymbol{\beta}}_+ \cdot \left(\sum_{u \in V_i}\boldsymbol{h}_u^{-1}\right)^{-1} + \hat{\boldsymbol{\beta}}'_+ \cdot \left(\sum_{u \in V_i}\boldsymbol{h}_u^{-1}\right)^{-1} \quad \forall i \in [n],$$

where $\hat{\boldsymbol{\beta}}_+, \hat{\boldsymbol{\beta}}'_+$ are the combined weights of $\boldsymbol{w}$'s in the positive direction, $\hat{\boldsymbol{\beta}}_-, \hat{\boldsymbol{\beta}}'_-$ are the combined weights of $\boldsymbol{w}$'s in the negative direction, and $c$ is a constant.

The above conditions can be satisfied with proper $\lambda_i$'s, so the model can fit all training data. Moreover, since the solution $\phi(G)^{\top}\boldsymbol{\beta}_{\mathrm{NTK}} = \left(\sum_{u \in V}\boldsymbol{h}_u^{-1}\right)^{-1}$ is equivalent to the functional form of the target function equation 7, GNN defined as in equation 5 can learn the `harmonic` task. □

Below, we prove GNNs with sum-aggregation and max-readout trained with squared loss in the NTK regime cannot extrapolate well on the `harmonic` task.

**Theorem 2.** Assume all nodes have the same scalar feature 1. Then, one-layer GNNs with sum-aggregation and max-readout trained with squared loss in the NTK regime do not extrapolate well in the `harmonic` task.

*Proof.* The target function of `harmonic` task is

$$f^{\star}(G) = \left(\sum_{v \in V}\left(\sum_{u \in N(v)} 1\right)^{-1}\right)^{-1},$$

so in order for one-layer GNNs with sum-aggregation and max-readout of the form

$$\text{MLP}\left(\max \sum_{v \in V} h_v\right), \quad h_v \text{ is the hidden vector for the node } v,$$

to match the target function, MLP must learn some non-linear transform between $\max$ and the inverse function. However, as shown in Xu et al. (2021), MLP converges to a linear function along directions from the origin. Hence, there always exist domains for which the GNN cannot learn the target function. $\qquad\square$

Similarly, we can show that one-layer GNNs with sum-aggregation and min/sum/mean-readout cannot learn the target function for some domain, meaning that they cannot extrapolate.

## B  TRAINING DETAILS

We used the open-source implementations of Set Transformer provided by the authors. We used the open-source implementation of SAGPool in Pytorch Geometric (Fey & Lenssen, 2019) provided by the authors with the reported hyperparameter settings. For all other models, we used the open-source implementations provided by the DGL framework (Wang et al., 2019).

For all models, we used the mean squared loss (MSE) as training and validation loss functions, unless otherwise stated. We performed a grid search to find the combination of hyperparameters that minimize the validation loss. In all experiments, we used the RMSprop optimizer (Tieleman & Hinton, 2012) to train all models with GNP, and for all baseline models, we additionally considered the Adam optimizer (Kingma & Ba, 2015) with default parameters (i.e., $\beta = (0.9, 0.999)$) and $\beta = (0.5, 0.999)$.

### B.1  EXTRAPOLATION ON GRAPH-LEVEL TASKS (RELATED TO SECTION 4.2)

For each task, we generated Erdős–Rényi (Erdős & Rényi, 1960) random graphs with probabilities ranging from 0.1 to 0.9. We trained and validated our model using such graphs with at least 20 and at most 30 nodes, and we tested on such graphs with at least 50 and at most 100 nodes, following the procedure in Xu et al. (2021). We generated $5,000$ graphs for training, $1,000$ graphs for validation, and $2,500$ graphs for test. For all nodes, we used the scalar 1 as the node feature.

For further experiments with different structures, we generated $2,500$ graphs of each type among `ladder` graphs, `4-regular` random graphs,[4] random `trees`, `expanders`,[5] and Barabási–Albert (`BA`) (Barabási & Albert, 1999) random graphs[6]. They all have at least 50 and at most 100 nodes.

Table 7 describes the hyperparameter search space for all graph-level tasks.

### B.2  EXTRAPOLATION ON NODE-LEVEL TASKS (RELATED TO SECTION 4.3)

We created $5,000$ graphs for training, $1,000$ graphs for validation, and $2,500$ graphs of each type for test in the way described in Section 4.2. Graphs for training and validation have at least 20 and at most 40 nodes, while those for test are larger with at least 50 and at most 70 nodes. Target nodes is sampled uniformly at random among all nodes in each graph.

For `shortest`, we used the scalar 0 as the feature of the target node, and used the scalar $10 \times |V|$ as the feature of the other nodes. The weight of each edge is drawn uniformly at random from $U(0, 5)$ in training and validation graphs, and from $U(0, 10)$ in test graphs. For `bfs`, we used the scalar 1 as the feature of the target node and used the scalar 0 as the feature for the other nodes. For both tasks, we added self-loop with edge weight 0 to every node.

Tables 8 describes the hyperparameter search space for all node-level tasks.

---

[4]The degree of every node is 4.

[5]We created Erdos-Renyi random graphs with probability 0.8, following the procedure in Xu et al. (2021).

[6]The number of edges to attach from a new node to existing nodes ranged from $0.05 \times |V|$ to $0.4 \times |V|$.

Table 7: Search space for `maxdegree`, `harmonic`, and `invsize` tasks

| Hyperparameter | Selection pool |
|---|---|
| Optimizer | RMSprop |
| Learning rate for $p$ | 3e-2, 1e-2, 3e-3 |
| Learning rate for the other parameters | 3e-2, 1e-2, 3e-3, 1e-3 |
| Norm clipping | 1e2, 1e4 |

(a) Search space for GIN with GNP

| Hyperparameter | Selection pool |
|---|---|
| Optimizer | Adam, Adam with $\beta = (0.5, 0.999)$, RMSprop |
| Learning rate | 3e-2, 1e-2, 3e-3, 1e-3 |
| Norm clipping | 1e2, 1e4 |
| Number of iterations (for Set2Set) | 1, 2 |

(b) Search space for GIN with baseline aggregation & readout functions

| Hyperparameter | Selection pool |
|---|---|
| Optimizer | Adam, Adam with $\beta = (0.5, 0.999)$, RMSprop |
| Learning rate | 1e-2, 3e-3, 1e-3, 3e-4 |
| Norm clipping | 1e2, 1e4 |

(c) Search space for SAGPool

Table 8: Search space for `bfs`, `shortest` tasks

| Hyperparameter | Selection pool |
|---|---|
| Optimizer | RMSprop |
| Learning rate for $p$ | 3e-2, 1e-2, 3e-3 |
| Learning rate for the other parameters | 1e-2, 3e-3, 1e-3 |
| Norm clipping | 1e2, 1e4 |

(a) Search space for GIN with GNP

| Hyperparameter | Selection pool |
|---|---|
| Optimizer | Adam, Adam with $\beta = (0.5, 0.999)$, RMSprop |
| Learning rate | 3e-2, 1e-2, 3e-3, 1e-3 |
| Norm clipping | 1e2, 1e4 |

(b) Search space for GIN with baseline aggregation functions

## B.3 EXTRAPOLATION ON SET-RELATED TASKS (RELATED TO SECTION 4.4)

For each task, we generated $4,000$ sets for training, $500$ sets for validation, and $500$ sets for test. For each set, the number of elements is sampled uniformly at random from $[20, 40)$ for training and validation sets, and from $[50, 100)$ for test sets. For $\mu_{\text{post}}$, $\sigma^2_{\text{post}}$, and $\hat{\mu}_{\text{MAP}}$, we sampled elements from $\mathcal{N}(\mu, 1^2)$ where $\mu \sim \mathcal{N}(0, 1^2)$. For $\hat{\sigma}^2_{\text{MAP}}$, we sampled elements $\mathcal{N}(5, \sigma^2)$ where $\sigma \sim \text{InvGamma}(1, 15)$. As loss functions, we used MSE for $\mu_{\text{post}}$ and $\sigma^2_{\text{post}}$ and used the negative logarithm of the product[7] of the likelihood and the prior for $\hat{\mu}_{\text{MAP}}$ and $\hat{\sigma}^2_{\text{MAP}}$.

Tables 9 describes the hyperparameter search space for all set-related tasks.

## B.4 GRAPH CLASSIFICATION (RELATED TO SECTION 4.7)

For original SAGPool (Lee et al., 2019b), we used the optimal hyperparameter settings shared by the authors[8]. For SAGPool equipped with GNP, we used gradient clipping with a maximum gradient norm of 1000 for the parameters of GNP, and we used a different learning rate for $p$ of GNP. For DD , PROTEINS, and NCI1, we used $1\times$, $20\times$, and $10\times$ larger learning rates for $p$ than the original learning rates, respectively.

---

[7] This product is proportional to the posterior probability

[8] https://docs.google.com/spreadsheets/d/1JXGNOCQkRHDCQqNarteYpEuWnkNzNq_WFiQrIY276i0/edit?usp=sharing

Table 9: Search space for $\mu_{\text{post}}$, $\sigma^2_{\text{post}}$, $\hat{\mu}_{\text{MAP}}$, and $\hat{\sigma}^2_{\text{MAP}}$

| Hyperparameter | Selection pool |
|---|---|
| Optimizer | RMSprop |
| Learning rate for $p$ | 3e-2, 1e-2, 3e-3 |
| Learning rate for the other parameters | 3e-2, 1e-2, 3e-3 |
| Norm clipping | 1e4 |

(a) Search space for GNP

| Hyperparameter | Selection pool |
|---|---|
| Optimizer | Adam, Adam with $\beta = (0.5, 0.999)$, RMSprop |
| Learning rate | 3e-2, 1e-2, 3e-3, 1e-3 |
| Norm clipping | 1e4 |

(b) Search space for basic operators

| Hyperparameter | Selection pool |
|---|---|
| Optimizer | Adam, Adam with $\beta = (0.5, 0.999)$, RMSprop |
| Learning rate | 1e-2, 1e-3, 1e-4 |
| Norm clipping | 1e4 |
| Number of iterations (for Set2Set) | 1, 2 |
| Encoder design (for Set transformer) | 2 SAB blocks, 2 ISAB blocks |

(c) Search space for Set2Set and Set Transformer

Table 10: Statistics of real-world datasets

| Dataset | Number of graphs | Average number of nodes | Average number of edges |
|---|---|---|---|
| D&D | 1178 | 284.3 | 715.7 |
| PROTEINS | 1113 | 39.06 | 72.82 |
| NCI1 | 4110 | 29.87 | 32.30 |

For original ASAPool (Ranjan et al., 2020), we used the optimal hyperparameter settings shared by the authors[9]. For ASAPool equipped with GNP, we used a different learning rate for $p$ and $q$ of GNP. For DD and NCI1, we used 3e-2 and 3e-3 for the learning rate for $p$ and $q$, respectively. For PROTEINS, we used 1e-1 for the learning rate for $p$, and 1e-2 for the learning rate for $q$.

### B.5 Influence Maximization (Related to Section 4.7)

For original MONSTOR (Ko et al., 2020), we used the optimal hyperparameter settings provided in the paper. For the parameters of GNP, we used the RMSprop optimizer, and the learning rates were set to 3e-2 for $p$ and 3e-3 for $q$.

## C  Additional Experiments and Results

### C.1  Graph-level Extrapolation on Real-world Datasets (Related to Section 4.2)

We further tested the extrapolation performances of GNP and baseline approaches using real-world graphs. For real-world graphs, we used D&D, PROTEINS, and NCI1, which were also used for graph classification tasks in the paper. Table 10 describes statistics of datasets. For evaluation, we ignored graphs with nodes with zero in-degrees.

In this experiment, we used a model trained using the Erdos–Rényi graphs described in Section 4.2. As seen in the Table 11, GNP showed near-perfect extrapolation performance only except for the maxdegree task on the NCI1 dataset. Even though the average number of nodes in the D&D dataset is approximately 10 times larger than that of the training dataset, the models trained with GNP performed well. One of the possible reasons for the relatively high MAPE on the NCI1 dataset is its extremely low average degree of nodes, which is roughly 2.17. Note that the training dataset contains Erdos–Rényi random graphs with edge probabilities ranging from 0.1 to 0.9.

[9] https://github.com/malllabiisc/ASAP

Table 11: Extrapolation performances of GNP on real-world datasets. Only except for the maxdegree task on the `NCI1` dataset, GNP showed near-perfect performance.

| Task | D&D | PROTEINS | NCI1 |
|---|---|---|---|
| invsize | 1.7±0.6 | 0.5±0.2 | 0.3±0.1 |
| harmonic | 3.4±1.1 | 2.3±0.3 | 2.4±0.7 |
| maxdegree | 3.4±1.3 | 2.8±1.1 | 22.4±12.4 |

Table 12: Extrapolation performance of baseline approaches on real-world datasets. Except for the `maxdegree` task, there was no combination of simple pooling functions that extrapolated well.

| Task | D&D | PROTEINS | NCI1 |
|---|---|---|---|
| invsize (best combination) | 100.0±0.0 (SortPooling) | 93.6±2.8 (SAGPool) | 37.9±0.0 (set2set) |
| harmonic (best combination) | 552.7±1012.4 (sum, mean) | 110.2±3.6 (sum, max) | 39.5±1.5 (sum, max) |
| maxdegree (ideal combination) | 0.0±0.0 (sum, max) | 0.0±0.0 (sum, max) | 0.0±0.0 (sum, max) |
| maxdegree (2nd best combination) | 10.5±0.9 (sum, mean) | 30.8±3.9 (sum, mean) | 63.9±15.0 (sum, mean) |

We also measured the test error of the baseline approaches, and we reported the test MAPE of the best-performing one in Table 12. Except for the `maxdegree` task, there was no combination of simple pooling functions that extrapolated well. These results are consistent with the experiment results in the paper. On the `maxdegree` task, the second best combination (among the 16 combinations of `sum`, `max`, `mean`, and `min`) showed significantly worse extrapolation performance than the GIN model equipped with `GNP`.

## C.2 GRAPH-LEVEL EXTRAPOLATION ON GRAPHS WITH DIFFERENT STRUCTURE TYPES (RELATED TO SECTION 4.2)

We trained a GNN using graphs of one structure type at a time then measured extrapolation error on the other structure types. In Table 13, each row denotes the test MAPEs of the model trained using the same graph. While the model trained using `ER` graphs or `BA` graphs extrapolated well on all three tasks, the model trained using the other graphs showed poor extrapolation performance. According to Xu et al. (2021), the distribution of training graphs can affect the extrapolation performance, and this can be one of the possible reasons why the model trained using `4regular`, `expander`, `tree`, `ladder` graphs showed poor extrapolation performance.

## C.3 GRAPH-LEVEL EXTRAPOLATION ON GRAPHS WITH DIFFERENT NODE FEATURE DISTRIBUTIONS (RELATED TO SECTION 4.2)

In the paper, we investigated the extrapolation performances in graph-level and node-level tasks on graphs with different sizes and structures. We also performed experiments on graphs with different edge feature distributions for the `shortest` task.

We additionally performed graph-level experiments for testing extrapolation to out-of-distribution node features. As in Xu et al. (2021), 3-dimensional node features drawn from $U(0, 5)$ were used in training and validation data, and those drawn from $U(0, 10)$ were used in test data.

We reported the test error in Table 14. As shown in the table, the error was slightly larger than that in the original settings without node features. However, the error was still reasonably low, and GNP outperformed baseline approaches especially on the `invsize` and `harmonic` tasks.

## C.4 GRAPH-LEVEL EXTRAPOLATION WITH VARIOUS ACTIVATION FUNCTIONS (RELATED TO SECTION 4.2)

We performed an additional graph-level experiment with a variant of GNP for handling negative inputs and a wider range of activation functions. Since the original GNP can only take non-negative inputs, we replaced ReLU to the absolute function for processing the inputs and then used an activation function. We considered ReLU, ELU, and LeakyReLU as the activation function. We compared the extrapolation error in each setting in Table 15, and GNP with the aforementioned changes showed performance comparable to original GNP.

Table 13: Test error on heterogeneous structures. Each row denotes the test MAPEs of the model trained using the same graph.

| | ER | BA | 4regular | Expander | Tree | Ladder |
|---|---|---|---|---|---|---|
| ER | 1.2±0.3 | 0.9±0.3 | 0.8±0.3 | 1.9±1.0 | 0.8±0.3 | 0.8±0.3 |
| BA | 1.4±1.3 | 1.1±0.7 | 1.1±0.5 | 1.9±2.7 | 1.0±0.5 | 1.1±0.5 |
| 4regular | 9.5±13.7 | 6.0±8.6 | 0.6±0.4 | 15.8±22.9 | 0.8±0.2 | 0.7±0.3 |
| Expander | 1.2±0.4 | 1.3±0.4 | 1.9±1.8 | 1.0±0.5 | 5.8±7.3 | 2.8±2.8 |
| Tree | 6.1±5.9 | 3.7±3.7 | 0.9±0.4 | 11.2±10.5 | 0.9±0.4 | 0.9±0.4 |
| Ladder | 5.1±2.7 | 3.1±1.6 | 1.4±0.5 | 7.0±5.1 | 2.7±3.0 | 1.1±0.6 |

(a) `invsize`

| | ER | BA | 4regular | Expander | Tree | Ladder |
|---|---|---|---|---|---|---|
| ER | 1.1±0.8 | 2.5±0.9 | 1.9±1.3 | 0.9±0.5 | 14.7±6.3 | 2.4±2.4 |
| BA | 8.0±2.2 | 2.9±0.5 | 2.7±0.6 | 13.7±4.0 | 7.2±3.1 | 2.8±2.0 |
| 4regular | 80.4±0.8 | 62.1±0.8 | 1.5±1.6 | 92.5±0.7 | 165.3±19.2 | 39.2±5.2 |
| Expander | 162.3±62.2 | 382.4±152.9 | 1202.6±526.6 | 7.8±2.2 | 3218.2±1491.6 | 1658.4±741.2 |
| Tree | 119.5±57.8 | 116.4±66.7 | 66.7±22.5 | 126.0±61.7 | 4.1±5.0 | 45.0±15.1 |
| Ladder | 85.8±0.4 | 72.4±0.6 | 24.8±0.9 | 94.8±0.2 | 99.8±7.3 | 4.0±0.9 |

(b) `harmonic`

| | ER | BA | 4regular | Expander | Tree | Ladder |
|---|---|---|---|---|---|---|
| ER | 2.5±0.4 | 2.1±1.1 | 3.4±3.7 | 2.3±1.1 | 1.9±0.6 | 30.7±16.9 |
| BA | 3.9±2.0 | 2.0±0.8 | 4.7±5.6 | 3.6±1.4 | 2.1±1.6 | 15.9±20.1 |
| 4regular | 86.7±3.5 | 89.2±2.1 | 5.0±8.1 | 92.3±3.3 | 15.5±1.5 | 38.8±8.5 |
| Expander | 13.2±9.2 | 23.5±4.6 | 238.6±138.7 | 7.4±2.6 | 131.6±96.7 | 306.7±174.7 |
| Tree | 78.8±28.2 | 57.7±21.3 | 41.3±13.2 | 126.9±44.7 | 2.6±0.5 | 31.0±22.2 |
| Ladder | 91.2±0.2 | 92.6±0.1 | 24.1±1.3 | 95.4±0.0 | 35.3±1.1 | 2.1±1.5 |

(c) `maxdegree`

Table 14: Test error on graph-level tasks with different node feature distributions.

| Task | invsize | harmonic | maxdegree |
|---|---|---|---|
| Test MAPE | 0.8±0.6 | 2.4±1.7 | 4.7±1.4 |

## C.5 TEST MAPE ON GRAPH-LEVEL TASKS (RELATED TO SECTION 4.2)

In Table 16, we reported test MAPEs and standard deviations for all 19 competitors and GNP on the graph-level tasks.

## C.6 BEHAVIORS OF GNP$^+$ AND GNP$^-$ FOR GRAPH-LEVEL TASKS (RELATED TO SECTION 4.5)

We analyzed the behavior of the negative GNP on three graph-level tasks that we performed in the paper. In all experiments, we found that either GNP$^+$ or GNP$^-$ tends to dominate the other side. To validate the observation, we masked the output of GNP$^+$ and GNP$^-$ for readout to 0 on the graph-level tasks.

As seen in Table 17, masking the output of GNP$^-$ on the maxdegree task and masking the output of GNP$^+$ on the other tasks do not significantly affect the extrapolation performance. When we masked the opposite part of GNP, however, the test MAPE was near 100. These results imply that the effect of the dominated part on the output of the model is negligible. That is, when the optimal pooling function is max, the negative GNP has almost no effect on determining the output. Similarly, when the optimal function is GNP with $(p, q) = (-1, 0)$, the positive GNP has almost no effect on determining the output.

## C.7 EFFECTIVENESS OF GNP ON NODE CLASSIFICATION AND GRAPH REGRESSION (RELATED TO SECTION 4.7)

**Node classification** To show the scalability of GNP to million-scale graphs, we performed node classification on the `OGBN-Arxiv` dataset (Hu et al., 2020), which contains 169, 343 nodes and 1, 166, 243 edges. We compared the node classification accuracy of GCN (Kipf & Welling, 2017)

Table 15: Test error on graph-level tasks with different activation functions.

| Task | invsize | harmonic | maxdegree |
|---|---|---|---|
| ReLU | 0.7±0.5 | 5.1±0.9 | 7.3±2.1 |
| LeakyReLU | 0.3±0.2 | 4.6±1.4 | 5.4±1.6 |
| ELU | 0.2±0.2 | 5.8±0.7 | 6.2±3.1 |

Table 16: Extrapolation performances on three graph-level tasks. We reported test MAPEs and standard deviations. Near-perfect scores are in bold, and scores significantly better than those in completely failed cases are underlined.

| Readout | Aggregation | | | | Readout | Test MAPE |
|---|---|---|---|---|---|---|
| | sum | max | mean | min | | |
| sum | 376.1±378.0 | 257.1±351.3 | 257.1±351.3 | 257.1±351.3 | SortPool | 100.0±0.0 |
| max | 101.0±7.6 | 179.1±44.2 | 179.1±44.2 | 179.1±44.2 | Set2Set | 198.8±0.5 |
| mean | 116.9±6.6 | 179.9±44.7 | 179.1±44.2 | 179.9±44.7 | SAGPool | 178.7±10.4 |
| min | 139.6±54.2 | 179.1±44.2 | 179.1±44.2 | 179.1±44.2 | **GNP** | **1.2±0.3** |

(a) `invsize`

| Readout | Aggregation | | | | Readout | Test MAPE |
|---|---|---|---|---|---|---|
| | sum | max | mean | min | | |
| sum | 109.6±19.0 | 121.1±28.9 | 151.5±73.1 | 121.1±28.9 | SortPool | 76.8±13.0 |
| max | 73.0±2.3 | 76.0±3.6 | 76.4±3.5 | 76.0±3.6 | Set2Set | 78.2±4.0 |
| mean | 95.7±9.5 | 75.9±3.6 | 76.3±3.5 | 75.9±3.6 | SAGPool | 26.9±21.0 |
| min | 91.1±12.2 | 76.0±3.6 | 76.3±3.5 | 76.0±3.6 | **GNP** | **1.1±0.8** |

(b) `harmonic`

| Readout | Aggregation | | | | Readout | Test MAPE |
|---|---|---|---|---|---|---|
| | sum | max | mean | min | | |
| sum | 60.5±22.1 | 50.5±2.1 | 49.9±0.5 | 50.5±2.1 | SortPool | **0.0±0.0** |
| max | **0.0±0.0** | 59.7±0.3 | 59.7±0.3 | 59.7±0.3 | Set2Set | **0.0±0.0** |
| mean | 16.3±2.4 | 59.7±0.3 | 59.7±0.3 | 59.7±0.3 | SAGPool | 51.4±1.8 |
| min | 25.5±3.2 | 59.7±0.3 | 59.7±0.3 | 59.7±0.3 | **GNP** | **2.5±0.4** |

(c) `maxdegree`

and GAT (Veličković et al., 2018), and their variants equipped with GNP. For implementations, we used open source implementations: GCN+linear+labels[10] and GAT+FLAG[11]. For the ones with GNP, since those implementations do not contain a linear layer after the last message-passing between the nodes, we used an additional linear layer only for the last GNN layer. The increase in the number of parameters is negligible compared to the total number of the parameters (see Table 18a). For GAT with GNP, we reduced the hidden dimension from 256 to 128.

We performed 10 runs for each model with the designated train/val/test split and reported the test accuracy and the standard deviation. As seen in Table 18a, GCN with GNP and GAT with GNP showed slightly better accuracy compared to the original models.

**Graph regression** We compared the graph regression performance of PNA (Corso et al., 2020) and its variant with GNP. We replaced all `max`, `sum`, `mean` and `min` aggregation in PNA with GNP. We used the `ZINC` (Irwin et al., 2012) dataset and performed experiments with and without edge features. For the implementation and the hyperparameters, we followed the implementation by the authors.[12]

We performed 5 runs for each model with the designated train/val/test split and reported the test MAE and the standard deviation in Table 18b. Without edge features, PNA equipped with GNP outperformed the original model. With edge features, however, the original model showed slightly lower test MAE than PNA equipped with GNP.

---

[10]https://github.com/dmlc/dgl/tree/master/examples/pytorch/ogb/ogbn-arxiv
[11]https://github.com/devnkong/FLAG
[12]https://github.com/lukecavabarrett/pna

Table 17: Test error with different masking schemes

| Task | without masking | masking GNP$^+$ | masking GNP$^-$ |
|------|-----------------|-----------------|-----------------|
| invsize | **1.2±0.3** | **1.1±0.1** | 99.6±1.0 |
| harmonic | **1.1±0.8** | **1.0±0.7** | 100.1±0.1 |
| maxdegree | **2.5±0.4** | 100.0±0.2 | **2.5±0.4** |

Table 18: Node classification and graph regression performance. The values in parentheses are the reported test accuracies/MAEs and the reported standard deviations.

| Model | Test accuracy | Number of parameters |
|-------|---------------|----------------------|
| GCN (original) | 0.7310±0.0014 (0.7306±0.0024) | 238,632 |
| **GCN (with GNP)** | **0.7324±0.0014** | **240,244** |
| GAT (original) | 0.7366±0.0017 (0.7371±0.0013) | 1,628,440 |
| **GAT (with GNP)** | **0.7377±0.0019** | **521,084** |

(a) Node classification results on `OGBN-Arxiv`

| Model | Test MAE | Number of parameters |
|-------|----------|----------------------|
| PNA (without edge features, original) | 0.3033±0.0116 (0.320±0.032) | 433,395 |
| **PNA (without edge features, with GNP)** | **0.2857±0.0360** | **433,443** |
| **PNA (with edge features, original)** | **0.1843±0.0090 (0.188±0.004)** | **95,111** |
| PNA (with edge features, with GNP) | 0.1910±0.0122 | 95,163 |

(b) Graph regression results on `ZINC`

# D    CLOSED-FORM SOLUTIONS FOR SET-RELATED TASKS (RELATED TO SECTION 4.4)

In Table 19, we provided the closed-form solutions for each task.

# E    DETAILS ABOUT INFLUENCE MAXIMIZATION AND MONSTOR (RELATED TO SECTION 4.7)

Influence Maximization (IM) (Kempe et al., 2003) is one of the most extensively studied NP-hard problems on social networks due to its practical applications in viral marketing and computational epidemiology. The goal of the problem is to choose a given number of seed nodes (i.e., a set of initially activated nodes) that maximize the influence through a given graph under a diffusion model. In this experiment, we used the Independent Cascade (IC) model as the diffusion model. In the IC model, each link $(u, v)$ has an activation probability $p_{uv}$. When a node $u$ is newly activated and a neighbor $v$ is not activated yet, the node $u$ has exactly one chance to activate the node $v$ with the probability $p_{uv}$, and the diffusion process ends when every activated node fails to activate any new node. In the model, the influence is the number of activated nodes after the diffusion process ends.

MONSTOR estimates the influence given a graph and a seed set. To train the model, we generated a dataset consisting of pairs of an input graph and a set of randomly chosen seed nodes. To generate ground-truth answers, we ran $10,000$ Monte-Carlo simulations and recorded the probability $\pi_{u,i}$ that each node $u$ is activated until the $i$-th step. We first trained the base model $M$ to estimate $\pi_i$ given $\pi_{i-1}, \ldots, \pi_{i-d}$. MONSTOR is constructed by stacking $s$ times the base model $M$, and $s$ is chosen to minimize squared loss between the ground-truth influences and the estimated influences on the validation set. Since influence maximization is a submodular maximization problem, we used UBLF (Zhou et al., 2013) or CELF (Leskovec et al., 2007) equipped with MONSTOR, which greedily selects seed nodes.

For the real-world datasets we used, we provided the statistics in Table 20.

Table 19: A closed-form solution for each task when the input set $S = \{x_1, x_2, \cdots, x_n\}$ is given.

| Task | Closed form solution |
|------|----------------------|
| $\mu_{\text{post}}$ | $\left(\frac{1}{\sigma_0^2} + \frac{n}{\sigma^2}\right)^{-1} \cdot \left(\frac{\mu_0}{\sigma_0^2} + \frac{1}{\sigma^2}\sum_{i=1}^n x_i\right)$ |
| $\sigma_{\text{post}}^2$ | $\left(\frac{1}{\sigma_0^2} + \frac{n}{\sigma^2}\right)^{-1}$ |
| $\hat{\mu}_{\text{MAP}}$ | $\left(\frac{1}{\sigma_0^2} + \frac{n}{\sigma^2}\right)^{-1} \cdot \left(\frac{\mu_0}{\sigma_0^2} + \frac{1}{\sigma^2}\sum_{i=1}^n x_i\right)$ |
| $\hat{\sigma}_{\text{MAP}}^2$ | $\left(\alpha + \frac{n}{2} + 1\right)^{-1} \cdot \left(\beta + \frac{1}{2}\sum_{i=1}^n (x_i - \mu)^2\right)$ |

Table 20: Statistics of real-world social networks used for the influence maximization task.

| Dataset | | Number of nodes | Number of edges |
|---------|-------|-----------------|-----------------|
| Extended | Train | 5636 | 31826 |
|          | Test  | 5413 | 27146 |
| Celebrity | Train | 7848 | 28839 |
|           | Test  | 7336 | 27699 |
| WannaCry | Train | 16246 | 84217 |
|          | Test  | 19381 | 85202 |

# F  CODE & DATA

All assets used in the paper, including the training/evaluation code and the trained models with GNP, are contained in the supplemental material. All assets we used from DGL[13] (Wang et al., 2019) and Pytorch Geometric[14] (Fey & Lenssen, 2019) are available under the Apache license 2.0 and MIT license, respectively. The implementation of Set Transformer[15] (Lee et al., 2019a) that we used is available under the MIT License. The implementation of ASAPool[16] (Ranjan et al., 2020) that we used is available under the Apache license 2.0. For the other assets, we were unable to find their licenses. For the SAGPool (Lee et al., 2019b) implementation in Pytorch Geometric, the dataset generators for the graph-level and node-level tasks (Xu et al., 2021), and the MONSTOR (Ko et al., 2020) implementation in DGL, we used the code on the GitHub repositories[17,18,19] shared by the authors of the original papers. We accessed TUDataset[20] (Morris et al., 2020) using PyTorch Geometric.

---

[13] https://github.com/dmlc/dgl
[14] https://github.com/rusty1s/pytorch_geometric
[15] https://github.com/juho-lee/set_transformer
[16] https://github.com/malllabiisc/ASAP
[17] https://github.com/inyeoplee77/SAGPool
[18] https://github.com/jinglingli/nn-extrapolate
[19] https://github.com/jihoonko/asonam20-monstor
[20] https://chrsmrrs.github.io/datasets/

