# OpenReview forum: "Learning to Pool in Graph Neural Networks for Extrapolation"
_ICLR.cc/2022/Conference — ICLR 2022 Submitted_

### Official Review · Reviewer_oxmV · 2021-10-30

**Correctness:** 3
**Technical Novelty And Significance:** 2
**Empirical Novelty And Significance:** 2
**Recommendation:** 5
**Confidence:** 4

**Main Review:**

The trainable pooling function eliminate the need to manually try different pooling setting. The pooling function performs steady and well on set-level tasks.

It appears challenging for GNP to learn the min or max pooling functions. Table 2 shows that GNP underperforms the minimum pooling by a large margin. Also, GNP performs no better than the maximum pooling on large graphs with heterogeneous structures. Hence, the proposed training is not able to push p to reach +/- infinity.

In section 3.2, instead of directly concatenating two vectors, why the output is obtained by taking the first half of GNP+ and the second half of GNP-. What does it mean by ‘let the model choose between them’? Under which kind of condition will the model choose p+ or p-?

The proposed method contains a one-layer MLP, which introduces extra d*(d + 1) parameters of the GNP functions compared to traditional pooling functions. The evaluation of using one-layer MLP is missing in the ablation study part. It’s hard to conclude whether the improvement comes from the MLP or the GNP.


**Summary Of The Paper:**

This paper presents Generalized Norm-based Pooling (GNP), a L^p norm like pooling function with the trainable p for Graph Neural Network to achieve extrapolation. The authors also propose a training method that use a single linear layer to address the gradient exploding issue. Equally splitting the GNP function, the model can learn both positive and negative p. Experiments were done to demonstrate GNP at the node-/graph-/set- levels.

**Summary Of The Review:**

The GNP idea seems promising. Additional clarifications and experiments may strengthen the manuscript.

---

> ### Author Response · Authors · 2021-11-23
> **Our Response**
>
> We appreciate your helpful comments.
>
> R4-1) It appears challenging for GNP to learn the min or max pooling functions. Table 2 shows that GNP underperforms the minimum pooling by a large margin. Also, GNP performs no better than the maximum pooling on large graphs with heterogeneous structures. Hence, the proposed training is not able to push p to reach +/- infinity.
>
> A4-1) In short, it is impossible to outperform “optimal” pooling functions (as in the mentioned task), and GNP performed best when all the tasks without optimal ones with only one exception. For details, please refer to A0-4 (in the general comment).
>
> R4-2) In section 3.2, instead of directly concatenating two vectors, why the output is obtained by taking the first half of GNP+ and the second half of GNP-. What does it mean by ‘let the model choose between them’? Under which kind of condition will the model choose p+ or p-?
>
> A4-2) We directly concatenate the outputs of GNP+ and GNP-. It does not mean that we only use half of the outputs. The weights of linear layers learn to choose the values in the concatenated vector. The model will choose p+ when the tasks are reasonable to learn with positive and p- in the opposite case.
>
>
> R4-3) The proposed method contains a one-layer MLP, which introduces extra d*(d + 1) parameters of the GNP functions compared to traditional pooling functions. The evaluation of using one-layer MLP is missing in the ablation study part. It’s hard to conclude whether the improvement comes from the MLP or the GNP.
>
> A4-3) In short, GNP does not require additional linear layers. For details, please refer to A0-1 (in the general comment).

---

> > ### Comment · Reviewer_oxmV · 2021-11-23
> > **Min/Max and Choosing p+ or p-**
> >
> > (a) In case of min/max, what are the values of "p" learned by GNP?
> > (b) Is it possible to take a peek at the weights to see if a layer clearly chooses p+ over p- or vice versa?

---

> > > ### Author Response · Authors · 2021-11-24
> > > **Our Response**
> > >
> > > (a) In case of min/max, what are the values of "p" learned by GNP?
> > >
> > > A-a) We demonstrated the empirical behavior of the values of "p" learned by GNP in Section 4.5. For details, please refer to Figure 2 and Section 4.5 (in the revised paper).
> > >
> > > (b) Is it possible to take a peek at the weights to see if a layer clearly chooses p+ over p- or vice versa?
> > >
> > > A-b) We also analyzed the behavior of the negative GNP on three graph-level tasks and found that either $GNP^+$ or $GNP^−$ tends to dominate the other side. For details, please refer to Appendix C.6 (in the revised paper).

---

> > > > ### Comment · Reviewer_oxmV · 2021-11-24
> > > > **Figure 2**
> > > >
> > > > In figure 2d, p- plateaus at 50, which is by design in page 4. Just curious, have you tried 100 or a higher bound? Does the choice of ε depends on the bound of p?

---

> > > > > ### Author Response · Authors · 2021-11-27
> > > > > **Our Response**
> > > > >
> > > > > Without the restriction, we observed that $p^-$ grew up to 147 during training. The bound of $p$ is insensitive to the value of $\epsilon$.

---

### Official Review · Reviewer_NEG6 · 2021-10-31

**Correctness:** 3
**Technical Novelty And Significance:** 1
**Empirical Novelty And Significance:** 2
**Recommendation:** 3
**Confidence:** 4

**Main Review:**

Endowing graph neural networks with extrapolation ability is a significant and challenging problem, as in many real cases the graph networks need to handle data within new environments during test time. This paper proposes to pursuit this goal via developing a new graph-level pooling function. The idea is very simple and straightforward and the proposed pooling function is a generalization of existing ones like max, sum, avg and min. Also, some theoretical analysis is provided to show that the new method can enable GNN for extrapolation in a simple example. To verify the approach, experiment results on nine different tasks, including graph-level, node-level and set-level, spanning from synthetic datasets to small real-world datasets, are conducted with comparison to existing models. The experiment designs are interesting and cover many diverse cases. However, there are some concerns as shown below.

The technical contribution of this paper is very weak. The proposed pooling function is a vector norm with the order coefficient as learnable parameters (i.e., only two for one pooling function). Then gradient method is considered for learning these parameters (4 in total) together with the GNN in an end-to-end manner. I doubt if this design could be applicable for large GNN models and would also suffer from serious optimization issue in practice. Although some practical tricks are proposed to avoid gradient explosion, it is not enough for guaranteeing a smooth training on large models and large real-world datasets. This concern is even amplified given the fact that the experiments mainly focus on synthetic datasets with unrealistic tasks and very small real-world datasets.

The theoretical analysis is also limited in a very constrained setting. The model learning is limited with square loss in NTK regime and the task is limited in harmonic task function. With these strong assumptions, it is not surprising to arrive at a proof of construction that satisfy the goal as shown in Thm 1, and such a result has little significance and impact for GNN learning on real tasks. Besides, in order to make the gradient smooth for optimization, the authors propose to combine the positive and negative p's in one formulation , however, such a design is not intuitive and hard to understand the rationale.

The experiment results are also weak. Most of the experiments are considered on synthetic datasets with unrealistic tasks that are tailored for the proposed function. A few results on real-world datasets are reported, however, these datasets are very small and the improvement achieved is not statistically significant. I suggest the authors add more experiments with some common graph benchmarks such as node classification/regression, link prediction/classification and graph classification/regression and consider more GNN backbones (GCN, GraphSAGE, GAT, etc.) to calibrate the comparison results. That could help to evaluate the practical efficacy of the propose method with respect to real applications.

**Summary Of The Paper:**

This paper focus on the extrapolation ability of graph neural networks and propose a new pooling function for graph-level readout based on vector norm. The proposed method can be applied to replace the commonly used pooling function like max/mean/sum in GNNs and is proved able for extrapolation in a simple example. Experiments on several synthetic tasks and a few real-world datasets are conducted to verify the effectiveness of the proposed method.

**Summary Of The Review:**

I carefully read the maintext and quickly go through the proof in the appendix. I am not sure about the correctness of the theoretical proof, though most of them appear to be correct. It is possible that I miss some details.

---

> ### Author Response · Authors · 2021-11-23
> **Our Response**
>
> We appreciate your helpful comments.
>
> R3-1) The technical contribution of this paper is very weak. The proposed pooling function is a vector norm with the order coefficient as learnable parameters (i.e., only two for one pooling function). Then gradient method is considered for learning these parameters (4 in total) together with the GNN in an end-to-end manner. I doubt if this design could be applicable for large GNN models and would also suffer from serious optimization issue in practice. Although some practical tricks are proposed to avoid gradient explosion, it is not enough for guaranteeing a smooth training on large models and large real-world datasets. This concern is even amplified given the fact that the experiments mainly focus on synthetic datasets with unrealistic tasks and very small real-world datasets.
>
> A3-1) In short, we used real-world graphs with up to **one million edges**. For details please refer to A0-3 (in the general comment).
>
> R3-2) The theoretical analysis is also limited in a very constrained setting. The model learning is limited with square loss in NTK regime and the task is limited in harmonic task function. With these strong assumptions, it is not surprising to arrive at a proof of construction that satisfy the goal as shown in Thm 1, and such a result has little significance and impact for GNN learning on real tasks. Besides, in order to make the gradient smooth for optimization, the authors propose to combine the positive and negative p's in one formulation, however, such a design is not intuitive and hard to understand the rationale.
>
> A3-2) As we described in the paper, we left the complete theoretical analysis as future work. Instead, we followed the overall proof process for one-layer GNN model with max/sum-readout on the maxdegree task described in Xu et al. 2021 and analyzed extrapolation ability in the NTK regime for one of the extrapolation tasks we performed.
> For the design combining the positive and negative p’s, we intuitively show that it is possible to learn to choose positive p or negative p in an end-to-end way. Also, we showed that the empirical behavior during training is consistent with the intuition for the ideal pooling function (See Section 4.5).
>
> R3-3) The experiment results are also weak. Most of the experiments are considered on synthetic datasets with unrealistic tasks that are tailored for the proposed function. A few results on real-world datasets are reported, however, these datasets are very small and the improvement achieved is not statistically significant. I suggest the authors add more experiments with some common graph benchmarks such as node classification/regression, link prediction/classification and graph classification/regression and consider more GNN backbones (GCN, GraphSAGE, GAT, etc.) to calibrate the comparison results. That could help to evaluate the practical efficacy of the propose method with respect to real applications.
>
> A3-3) In short, in the revised paper, we used seven GNN architectures and four real-world tasks (node classification, graph classification, graph regression, and influence maximization). For details, please refer to A0-3 and A0-5 (in the general comment).

---

> > ### Comment · Reviewer_NEG6 · 2021-11-30
> > **Thank you for the response**
> >
> > Thank you for the response and adding more experiment results. However, I would like to insist on the initial review. Here are the justifications. First, the intuition and rationale behind the design are still not clear to me given that the theoretical analysis is based on unrealistic assumptions. Second, the current empirical performance is not that encouraging and the improvement over baselines on large datasets is very small. Though I like the scope and topic of this work, I would suggest the authors improve the method and re-organize the experiment designs to make the technical contributions more convincing.

---

### Official Review · Reviewer_17q4 · 2021-11-02

**Correctness:** 3
**Technical Novelty And Significance:** 2
**Empirical Novelty And Significance:** 2
**Recommendation:** 3
**Confidence:** 4

**Main Review:**


**Strengths**
1. The paper is well written, with a clear description and proof of the GNP method.
2. The design of GNP is simple but general, enabling the model to learn which aggregation and pooling method to use.
3. The extrapolation performance of the GNP outperforms simple min, max, mean, and sum pooling methods.

**Weakness**
1. Though GNP can theoretically simulate the behavior of the max and min, I am concerning the $p\rightarrow \pm\infty$ may cause stability issues for computation. That might be the reason why GNP performs worse than (sum, max) on the max degree task in Figure 1(c).
2. The GNP requires the inputs to be non-negative so that the inputs should be the output of the ReLU function. It may cause information loss of the embeddings. It is a common practice in machine learning that the inputs of the pooling function should not be the output of the non-linear function. Therefore, it would be more convincing if the authors can show results on more real-world datasets, e.g. the ogb dataset.
3. The baseline results on the NCI dataset shown in the paper is not SOTA, as in the original paper $\text{SAGPool}_g$ achieves 0.742. Also, there are stronger baselines on that dataset, including [1, 2] which achieve 0.815.
4. GNP is an interesting idea and performs well on small synthetic datasets, but it would be better to see how GNP works on state-of-the-art model architectures on real-world applications, like PNA[3] and DAGNN[4], instead of the vanilla GIN and GCN.
5. Node classification is another task that is good to see in the paper to demonstrate the effectiveness of using the GNP for aggregation.

[1] U. Alon, and E. Yahav. On the Bottleneck of Graph Neural Networks and its Practical Implications. ICLR 2021.

[2] F. Errica, M. Podda, D. Bacciu, and A. Micheli. A fair comparison of graph neural networks for graph classification. ICLR 2020.

[3] Gabriele Corso, Luca Cavalleri, Dominique Beaini, Pietro Liò, and Petar Veličković. Principal Neighbourhood Aggregation for Graph Nets. NeurIPS 2020.

[4] V. Thost, and J. Chen. Directed Acyclic Graph Neural Networks. ICLR 2021.

**Summary Of The Paper:**

The paper proposes a learnable pooling method (GNP) for GNN based on $L^p$ norm, which is general to simulate the behavior of max, min, sum, and average. It performs well on many node-level, graph-level, and set-related tasks.

**Summary Of The Review:**

The GNP idea proposed in the paper is interesting and works well on small tasks. However, the non-negative constraint of the inputs may be too strong and can cause information loss. Without state-of-the-art large-scale real-world experimental results, I am not fully convinced by the effectiveness of GNP.

---

> ### Author Response · Authors · 2021-11-23
> **Our Response**
>
> We appreciate your helpful comments.
>
> R2-1) Though GNP can theoretically simulate the behavior of the max and min, I am concerning the  p→±∞ may cause stability issues for computation. That might be the reason why GNP performs worse than (sum, max) on the max degree task in Figure 1(c).
>
> A2-1) In short, it is impossible to outperform “optimal” pooling functions (as in the mentioned task), and GNP performed best when all the tasks without optimal ones with only one exception. For details, please refer to A0-4 (in the general comment).
>
> R2-2) The GNP requires the inputs to be non-negative so that the inputs should be the output of the ReLU function. It may cause information loss of the embeddings. It is a common practice in machine learning that the inputs of the pooling function should not be the output of the non-linear function. Therefore, it would be more convincing if the authors can show results on more real-world datasets, e.g. the ogb dataset.
>
> A2-2) We would like to emphasize that we used real-world graphs with up to **85,000 edges** for the Influence maximization task. Also, in the revised paper, we used a graph with over **one million edges** to demonstrate the effectiveness of GNP on larger real-world graphs. For details, please refer to A0-3 (in the general comment).
>
>
> R2-3) The baseline results on the NCI dataset shown in the paper is not SOTA, as in the original paper SAGPoolg achieves 0.742. Also, there are stronger baselines on that dataset, including [1, 2] which achieve 0.815.
>
> A2-3) Our response is two-fold:
> - In our graph classification experiments, we compared the performance with the hierarchical SAGPool (SAGPool_h) and ASAPool, and measured test accuracy of their variants with GNP.
> - Since we focus on the performance gain when GNP is applied to **existing** GNN architectures, we do not consider SAGPool_g and any advanced architectures as direct competitors of GNP. Note that GNP can be applied to various GNN architectures, including what the reviewer mentioned, as described in A0-2 (see the general comment).
>
>
> R2-4) GNP is an interesting idea and performs well on small synthetic datasets, but it would be better to see how GNP works on state-of-the-art model architectures on real-world applications, like PNA[3] and DAGNN[4], instead of the vanilla GIN and GCN.
> A2-4) Our response is two-fold:
> - In short, we applied GNN to seven different GNN architectures. For details, please refer to A0-2 (in the general comment).
> - In the revised paper, we applied GNP to PNA on a graph regression task, as described in A0-5 (see the general comment).
>
> R2-5) Node classification is another task that is good to see in the paper to demonstrate the effectiveness of using the GNP for aggregation.
>
> A2-5) In the revised paper, in order to demonstrate the effectiveness of GNP on larger real-world graphs, we used GNP for node classification, as described in A0-3 (see the general comment).

---

> > ### Comment · Reviewer_17q4 · 2021-11-30
> > **Thank you for the responses**
> >
> > I appreciate the authors' detailed responses. It is great to see the results on the ogbn-arxiv dataset, and the proposed method is slightly better than the vanilla GCN and GAT. I am still concerned about the real effectiveness of the proposed GNP. I am wondering if there is a real large-scale problem that the GNP will outperform the existing methods. Or it would be great to show that GNP has a more stable performance on multiple practical tasks compared to the existing work, as the authors claim that the proposed method is more general. It would be better to investigate the proposed method more thoroughly, thus I would like to maintain my original score.

---

### Official Review · Reviewer_hW8Y · 2021-11-07

**Correctness:** 3
**Technical Novelty And Significance:** 2
**Empirical Novelty And Significance:** 2
**Recommendation:** 5
**Confidence:** 3

**Main Review:**

The authors propose GNP, a new pooling operation for GNNs, which subsumes max, min, mean and sum as special cases.

The contribution to this paper is incremental. It builds on top of the well known L-p norm pooing function and extends it to allow negative values of p, and an additional learnable parameter q. However, this simple extension is not a well behaved function for gradient based optimization, hence the authors tweak the layer to behave better using techniques that are well known.

Based on the experimental results, it does look like GNP outperforms the baselines which is a plus. However, in some cases, despite being a more complex pooling layer with more learning parameters than the baseline pooling functions, it does not seem to offer enough gain in performance to justify its complexity. (Especially since in some of the experiments, the optimal baseline was part of the search space for GNP). For example:
Table 1: GNP is unable to match with (sum, max) on the maxdegree task for Ladder Graphs.
Table 2: GNP is unable to match min on bfs.
This is disappointing considering that GNP has the upper hand in the # of learnable parameters

The real world graphs used in the paper comprise a total of ~6k graphs. The heaviest of this dataset, D&D contains ~300 nodes on average. While GNP performs well on these graphs (except for maxdegree), I feel the datasets are not large enough to assert if GNP scales to very large graphs containing hundreds of thousands of nodes. Further, the synthetic datasets used contain only 50-100 nodes, and hence are also relatively small.

The ablation study was interesting. It is nice to see that in some cases GNP is able to find the optimum values of p and q, especially for maxdegree. For the GNP+ part of the ablation, do you still use the extra linear layer described in 3.2? I am curious about how the GNP+ model fairs with and without that extra linear layer.



**Summary Of The Paper:**

Authors propose GNP, a new pooling operation for Graph Neural Network which subsumes max, min, mean and sum as special cases.

The premise of the paper is promising - if a single layer can provide the performance of max, min mean and sum, through two new learning parameters, it would be a big win for research. On certain occasions, it is unclear which pooling function is the best and researchers resort to large grid searches to identify which ones work best. This could do away with that.

However, I feel that the paper falls a little short of delivering this.

**Summary Of The Review:**

For an excellent version of this paper, I would recommend the following:
- Evaluate the models on more real world datasets with larger graphs - 100+ and 1000+ nodes and break down the performance of GNP based on the number of nodes in the graph. It is often the case that GNN architectures don't scale well on larger graphs.
- Add an ablation study with GNP+ with and without the additional linear layer.

---

> ### Author Response · Authors · 2021-11-23
> **Our Response**
>
> We appreciate your helpful comments.
>
> R1-1) In some cases, despite being a more complex pooling layer with more learning parameters than the baseline pooling functions, it does not seem to offer enough gain in performance to justify its complexity. (Especially since in some of the experiments, the optimal baseline was part of the search space for GNP). For example: Table 1: GNP is unable to match with (sum, max) on the maxdegree task for Ladder Graphs. Table 2: GNP is unable to match min on bfs. This is disappointing considering that GNP has the upper hand in the # of learnable parameters.
>
> A1-1) We described our response about the concerns in A0-4 (see the general comment).
>
> R1-2) The real world graphs used in the paper comprise a total of ~6k graphs. The heaviest of this dataset, D&D contains ~300 nodes on average. While GNP performs well on these graphs (except for maxdegree), I feel the datasets are not large enough to assert if GNP scales to very large graphs containing hundreds of thousands of nodes. Further, the synthetic datasets used contain only 50-100 nodes, and hence are also relatively small.
>
> A1-2) We would like to emphasize that we used real-world graphs with up to **85,000 edges** for the Influence maximization task. Also, in the revised paper, we used a graph with over **one million edges** to demonstrate the effectiveness of GNP on larger real-world graphs. For details, please refer to A0-3 (in the general comment).
>
> R1-3) It is nice to see that in some cases GNP is able to find the optimum values of p and q, especially for maxdegree. For the GNP+ part of the ablation, do you still use the extra linear layer described in 3.2? I am curious about how the GNP+ model fairs with and without that extra linear layer.
>
> A1-3) In short, GNP does not require additional linear layers. For details, please refer to A0-1 (in the general comment).

---

> > ### Comment · Reviewer_hW8Y · 2021-11-29
> > **Response**
> >
> > Thank you for the detailed response. I have gone through the review and acknowledge that the authors have evaluated the technique on larger, real world graphs. However, I maintain my initial position that this technique is more complicated due to the complex gradient profile of the new layer and at the same time only offers marginal improvements over existing architectures. Due to this, I believe that most practitioners will prefer the vanilla layers.

---

### Author Response · Authors · 2021-11-18
**General Response**

For all reviewers,
We’d like to thank the reviewers for helpful comments. However, we found that there are some common misunderstandings about our paper. We provide a general response to all reviewers about the misunderstandings. The paper was clarified to prevent the misunderstanding. For the rest of the questions and concerns, we responded to individual reviewers.

R0-1) Does GNP have a higher complexity or more parameters than baselines?

A0-1) We have only **four** extra parameters ($p^+$, $p^-$, $q^+$ and $q^-$) for each layer, compared to other widely-used GNN layers such as GCN and GIN. Note that they also have a linear layer or MLP after message-passing between nodes. Instead of using an additional linear layer, GNP concatenates the outputs of $GNP^+$ and $GNP^-$ and passes them to the linear layer or MLP. Also, note that we set the dimensions so that the concatenated output of GNP has the same dimension as the output of other GNNs. These facts are clarified in the revised paper.

R0-2) I suggest the authors consider more GNN backbones.

A0-2) We would like to remind the reviewers that we applied GNP to **five** GNN architectures in the original submission (and additionally 2 in the revised paper). For aggregation, we successfully applied GNP to GIN for extrapolation experiments, and to GCN for graph classification and MONSTOR for influence maximization. We also showed that ASAPool and hierarchical SAGPool equipped with GNP consistently outperformed the original models with carefully chosen pooling functions on the task of graph classification. In the revised paper, we additionally used GAT and PNA for node classification and graph regression, respectively.

R0-3) Is there any experiment on large real-world graphs (graphs with more than 100+ and 1000+ nodes)?

A0-3) Our response is three-fold:
Most real-world graphs contained in the common graph classification benchmark have similar sizes to the ones we experimented with. For example, the average number of nodes of the largest dataset for OGB graph classification benchmark is 243.4.
We performed the Influence maximization experiment and outperformed the baselines on real-world graphs with up to over **85,000** edges (see Section 4.7). The table below shows the size of the considered real-world graphs:

| Datasets  | Number of nodes               | Number of edges               |
|-----------|-------------------------------|-------------------------------|
| Extended  | 5636 (Train) / 5413 (Test)   | 31826 (Train) /  27146 (Test) |
| Celebrity | 7848 (Train) /  7336 (Test)   | 28839 (Train) / 27699 (Test)  |
| WannaCry  | 16246 (Train) /  19381 (Test) | 84217 (Train) / 85202 (Test)  |

In the revised paper, we show the scalability of GNP, we performed node classification on the OGBN-Arxiv dataset ($|V| = 169,343$, $|E|=1,166,243$). We compared the node classification accuracy of GCN and GAT, and their variants equipped with GNP. For implementations, we used open source implementations: GCN+linear+labels (https://github.com/dmlc/dgl/tree/master/examples/pytorch/ogb/ogbn-arxiv) and GAT+FLAG (https://github.com/devnkong/FLAG). For the ones with GNP, since those implementations do not contain a linear layer after the last message-passing between the nodes, we used an additional linear layer only for the last GNN layer. The increase in the number of parameters is negligible compared to the total number of the parameters (see the table below). For GAT with GNP, we reduced the hidden dimension from 256 to 128. We performed 10 runs for each model with the designated train/val/test split and reported the test accuracy and the standard deviation. As seen in the table, GCN with GNP and GAT with GNP showed slightly better accuracy compared to the original models.

|Model|Test accuracy|Number of parameters|
|-|-|-|
| GCN (original) | 0.7310 ± 0.0014 (Reported: 0.7306 ± 0.0024) | 238,632 |
| GCN (with GNP) | 0.7324 ± 0.0014 | 238,632 |
| GAT (original) | 0.7366 ± 0.0017 (Reported: 0.7371 ± 0.0013) | 1,628,440 |
| GAT (with GNP) | 0.7377 ± 0.0019 | 521,084 |

---

> ### Author Response · Authors · 2021-11-23
> **General Response (Continued)**
>
> R0-4) The performances of GNP decrease in some cases (e.g.  GNP vs (sum, max) on the maxdegree task in Table 1, GNP vs min on the shortest path task in Table 2.(b)), although the number of parameters increases.
>
> A0-4) As mentioned in A0-1, GNP only has 4 more parameters per layer compared to other well-known GNN models. On the shortest path task, min is the “**optimal**” pooling function, which can output exact solutions. Thus, it is impossible to outperform it, while GNP outperformed all other baselines. Also, GNP performed **best** on all tasks where no optimal aggregation or pooling functions are known (see Table 5 and Table 6) with only one exception.
>
> R0-5) It would be better to see how GNP works on state-of-the-art model architectures on real-world applications.
>
> A0-5) We compared the graph regression performance of PNA and its variant with GNP. We replaced all max, sum, mean and min aggregation with GNP. We used the ZINC dataset and performed experiments with and without edge features. For the implementation and the hyperparameters, we followed the open-source implementation by authors (https://github.com/lukecavabarrett/pna). We performed 5 runs for each model with the designated train/val/test split and reported the test MAE and the standard deviation. Without edge features, PNA equipped with GNP outperformed the original model. With edge features, the original model showed slightly lower test MAE than PNA equipped with GNP.
>
> |Model|Test MAE|Number of parameters|
> |-|-|-|
> | PNA (without edge features, original) | 0.3033 ± 0.0116 (Reported: 0.320 ± 0.032) | 433,395 |
> | PNA (without edge features, with GNP) | 0.2857 ± 0.0360 | 433,443 |
> | PNA (with edge features, original) | 0.1843 ± 0.0090 (Reported: 0.188 ± 0.004) | 95,111 |
> | PNA (with edge features, with GNP) | 0.1910 ± 0.0122 | 95,163 |

---

### Author Response · Authors · 2021-11-30
**General Response**

We would like to emphasize that our main contribution is **extrapolation**, rather than achieving the highest score in benchmarking tasks unrelated to extrapolation. In this work, we verified that simply replacing pooling functions with GNP enables GNNs to extrapolate well on many node-level, graph-level, and set-related tasks, while they can fail with conventional pooling functions even on small datasets.

---

### Decision · Program_Chairs · 2022-01-20

**Decision:**

Reject

**Comment:**

This paper focuses on the extrapolation ability of graph neural networks and proposes a new pooling function based on vector norm. The proposed method can be applied to replace the commonly used pooling function like max/mean/sum, and is proved able for extrapolation in a simple example.

Overall, all reviewers tend to reject this submission due to the following reasons
- The contribution to this paper is incremental. It builds on top of the well-known L-p norm pooing function and extends it to allow negative values of p and an additional learnable parameter q.
- However, this simple extension is not a well-behaved function for gradient-based optimization, which leads to unconvinced experiments, i.e., diverse performance compared with min/max.
- more recent baselines should be compared with and it would be better to see how GNP works on state-of-the-art model architectures on real-world applications